# Structure-based inhibitors of amyloid beta core suggest a common interface with tau

Sarah L Griner[1]*, Paul Seidler[1], Jeannette Bowler[1], Kevin A Murray[1], Tianxiao Peter Yang[1], Shruti Sahay[1], Michael R Sawaya[1], Duilio Cascio[1], Jose A Rodriguez[1], Stephan Philipp[2], Justyna Sosna[2], Charles G Glabe[2,3], Tamir Gonen[4†], David S Eisenberg[1]*

[1]UCLA-DOE Institute, Department of Biological Chemistry, Molecular Biology Institute, Howard Hughes Medical Institute, University of California, Los Angeles, Los Angeles, United States; [2]Department of Molecular Biology and Biochemistry, University of California, Irvine, Irvine, United States; [3]Biochemistry Department, Faculty of Science and Experimental Biochemistry Unit, King Fahd Medical Research Center, King Abdulaziz University, Jeddah, Saudi Arabia; [4]Howard Hughes Medical Institute, Janelia Research Campus, Ashburn, United States

**Abstract** Alzheimer's disease (AD) pathology is characterized by plaques of amyloid beta (Aβ) and neurofibrillary tangles of tau. Aβ aggregation is thought to occur at early stages of the disease, and ultimately gives way to the formation of tau tangles which track with cognitive decline in humans. Here, we report the crystal structure of an Aβ core segment determined by MicroED and in it, note characteristics of both fibrillar and oligomeric structure. Using this structure, we designed peptide-based inhibitors that reduce Aβ aggregation and toxicity of already-aggregated species. Unexpectedly, we also found that these inhibitors reduce the efficiency of Aβ-mediated tau aggregation, and moreover reduce aggregation and self-seeding of tau fibrils. The ability of these inhibitors to interfere with both Aβ and tau seeds suggests these fibrils share a common epitope, and supports the hypothesis that cross-seeding is one mechanism by which amyloid is linked to tau aggregation and could promote cognitive decline.
DOI: https://doi.org/10.7554/eLife.46924.001

**\*For correspondence:**
sgriner@ucla.edu (SLG);
david@mbi.ucla.edu (DSE)

**Present address:** †Department of Physiology, David Geffen School of Medicine, University of California, Los Angeles, Los Angeles, United States

## Introduction

Although Alzheimer's disease (AD) is the most prevalent form of dementia, there are limited treatments to alleviate symptoms and none that halt its progression. Histological features of AD are extracellular senile plaques of amyloid beta (Aβ) and intracellular neurofibrillary tangles of tau (*Glenner et al., 1984*; *Goedert et al., 2017*). While Aβ aggregation is thought to occur at the early stages of AD, tau aggregation correlates better to disease progression, with characteristic spreading along linked brain areas, and severity of symptoms correlating to the number of observed inclusions (*Tanzi, 2012*; *Hardy and Selkoe, 2002*; *Manczak and Reddy, 2014*; *Seward et al., 2013*; *Brier et al., 2016*; *Schwarz et al., 2016*). Structural information about the aggregated forms of Aβ and tau is accumulating, but to date this knowledge has not led to successful chemical interventions (*Chen et al., 2017*).

A link between the appearance of Aβ and tau pathologies has been noted in transgenic mouse models generated by crossing or co-expressing mutant Aβ and mutant tau, but the mechanism is not yet understood at the molecular level (*Oddo et al., 2003*). By injecting Aβ seeds derived from synthetic peptide, transgenic mouse or AD patient tissue, tau pathology can be found both at the site of injection, and also in functionally connected brain areas (*Bolmont et al., 2007*; *Götz et al., 2001*; *Morales et al., 2015*). Tau aggregation has also been reported to follow Aβ seeding in 3D

neuronal stem cell cultures that express early onset hereditary mutations to drive overproduction and aggregation of Aβ (*Choi et al., 2014*). In spite of these observations, the molecular linkage of Aβ to tau remains unresolved. Proposed hypotheses include Aβ causing downstream cellular changes that trigger tau phosphorylation and eventual aggregation, and/or a direct interaction and seeding of tau by aggregated Aβ (*Ittner and Götz, 2011*; *Stancu et al., 2014*; *Morales et al., 2013*).

Several lines of evidence support the direct interaction model, although questions still remain; for example, how such an interaction could occur since Aβ plaques deposit extracellularly, while tau neurofibrillary tangles are intracellular. One possible model for intracellular aggregation could be that Aβ is cleaved from APP inside endosomes, and then exported (*Rajendran et al., 2006*). Another model proposes that smaller diffusible Aβ oligomers are the toxic species (*Lesné et al., 2006*; *Lambert et al., 1998*; *Benilova et al., 2012*); indeed oligomers of Aβ isolated from AD serum are sufficient to induce tau aggregation (*Jin et al., 2011*). Aβ has also been found co-localize intra-neuronally with tau as well as at synaptic terminals, with increased interactions correlating with disease progression (*Manczak and Reddy, 2014*). Furthermore, soluble and insoluble complexes of Aβ bound to tau have been detected in AD tissue extracts (*Manczak and Reddy, 2014*; *Guo et al., 2006*). In vitro, soluble complexes of Aβ and tau have been found to promote aggregation of tau (*Guo et al., 2006*), while another study found that Aβ fibrils can seed tau (*Vasconcelos et al., 2016*). Taking the evidence together, we hypothesize that cross-seeding of tau by Aβ promotes tangle formation in AD, which could be prevented not only by inhibiting Aβ aggregation, but also by disrupting the binding site of Aβ with tau.

A number of interaction sites have been proposed on both proteins. In Aβ, both the amyloid core KLVFFA, along with region spanning the carboxy terminal residues were found to bind tau (*Guo et al., 2006*). Conversely peptides from regions of tau in exons 7 and 9, well as aggregation prone sequences VQIINK and VQIVYK located at the beginning of repeat 2 (R2) and repeat 3 (R3) of the microtubule domain (K18), respectively, were found to bind Aβ (*Guo et al., 2006*). A computational seeding model predicts that the amyloid core of Aβ can form intermolecular β-sheet interactions with VQIINK or VQIVYK (*Miller et al., 2011*).

On this basis, we hypothesized that an inhibitor capable of targeting the amyloid core, which itself is an important sequence for Aβ aggregation (*Tjernberg et al., 1999*; *Bernstein et al., 2005*; *Marshall et al., 2016*), might block both Aβ aggregation and tau seeding by Aβ. However, this segment has been observed in multiple conformations in steric zipper structures (*Colletier et al., 2011*) and fiber models (*Lührs et al., 2005*; *Colvin et al., 2016*; *Qiang et al., 2012*; *Huber et al., 2015*; *Wälti et al., 2016*), impeding structure-based inhibitor design. In an effort to characterize a toxic conformation of this sequence, we focused our efforts on determining the structure of the segment 16–26, containing the Iowa early onset hereditary mutation, D23N (*Van Nostrand et al., 2001*). Based on this structure, we designed several inhibitors and found that they indeed blocked aggregation of Aβ, prevented cross-seeding of tau by Aβ, and surprisingly, also blocked tau homotypic seeding. We suggest that the efficacy of these structure-based inhibitors against both proteins, but not other amyloid fibrils, implies there is a similar binding interface displayed on both Aβ and tau aggregates, supporting the cross-amyloid cascade hypothesis in AD.

## Results

### Atomic structure of Aβ$_{16-26}$ D23N determined using MicroED

With crystals only a few hundred nanometers thick, we used micro-electron diffraction (MicroED) to determine the structure of Aβ residues 16–26 containing the hereditary mutation D23N, (*Figure 1A*), KLVFFAENVGS. The structure revealed pairs of anti-parallel β-sheets each composed of ~4000 strands, stacked into a fibril that spans the entire length of the crystal. Neighboring sheets are oriented face to back (*Figure 1B*, *Table 1*) defining a Class seven steric zipper motif.

In addition, the three C-terminal residues adopt an extended, non-β conformation which stabilizes the packing between steric zippers (*Figure 1—figure supplement 1*). The sheet–sheet interface is strengthened by interdigitating side chains, Lys 16, Val18, Phe20, Glu22 from the face of one strand, and Leu17, Phe19, and the N-terminus from the back of the other. The zipper has an extensive

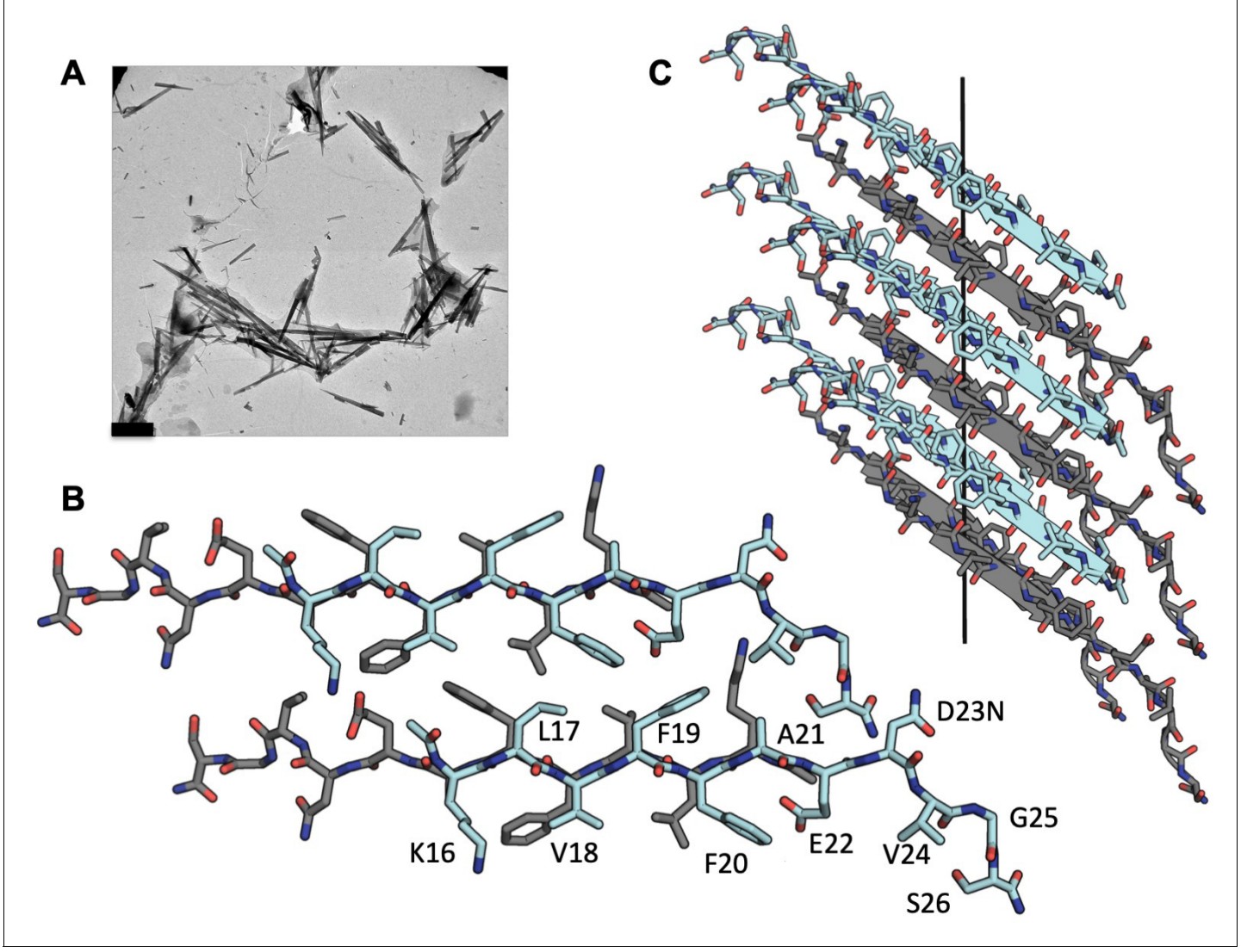

**Figure 1.** MicroED structure of segment Aβ 16–26 D23N from microcrystals. (**A**) Electron micrograph of 3D crystals used for data collection, scale bar is 1 µm. (**B**) The crystal structure reveals tightly mated pairs of anti-parallel β-sheets with opposing sheets in gray and cyan. The side-chains interdigitate to form a dry interface. Two neighboring sheets are viewed perpendicular to the β-sheets. (**C**) View of 6 layers perpendicular to the fibril axis (black line). The β-sheets stack out of register along the fibril axis.

DOI: https://doi.org/10.7554/eLife.46924.002

The following figure supplements are available for figure 1:

**Figure supplement 1.** Crystal packing of the Aβ 16–26 D23N atomic structure.
DOI: https://doi.org/10.7554/eLife.46924.003

**Figure supplement 2.** The spines of Aβ 16–26 D23N and Aβ$_{1-42}$ fibrils (5OQV) are structurally similar.
DOI: https://doi.org/10.7554/eLife.46924.004

interface with a high shape complementarity of 0.76 and a total buried solvent accessible surface area of 258 Å$^2$.

This structure is partly identical to that of a shorter peptide segment, Aβ16–21, KLVFFA (crystal form-I) (*Colletier et al., 2011*), which we used successfully as a search model for molecular replacement. Both the longer and shorter segments have class seven symmetry. However, the two segments differ in registry. The shorter segment maintains an in-register hydrogen bonding pattern while the longer segment is out-of-register. That is, the strands of Aβ16–26 are tilted away from perpendicular to the fibril axis—a departure from canonical cross-β architecture. This elongated beta

**Table 1.** Statistics of MicroED data collection and atomic refinement.

| | KLVFFAENVGS |
|---|---|
| Excitation Voltage (kV) | 200 |
| Electron Source | field emission gun |
| Wavelength (Å) | 0.0251 |
| Total dose per crystal ($e^-$/ Å$^2$) | 2.7 |
| Frame rate (frame/s) | 0.3–0.5 |
| Rotation rate (°/s) | 0.3 |
| #crystals used | 13 |
| Total angular rotation collected (°) | 941 |
| Merging Statistics | |
| Space group | P2$_1$ |
| Cell dimensions | |
| a, b, c (Å) | 11.67, 51.91, 12.76 |
| α, β, γ (°) | 90, 114.18, 90 |
| Resolution (Å) | 11.64–1.4 (1.44–1.40)[*] |
| $R_{merge}$ | 24.0% (65.2%) |
| No. Reflections | 47,598 (1966) |
| Unique Reflections | 2355 (163) |
| Completeness (%) | 86.2% (78.0%) |
| Multiplicity | 21 (12) |
| I/σ | 9.06 (2.88) |
| $CC_{1/2}$ | 99.5% (69.7%) |
| Refinement Statistics | |
| No. reflections | 2354 |
| Reflections in test set | 236 |
| $R_{work}$ | 23.7% |
| $R_{free}$ | 28.3% |
| R.m.s. deviations | |
| Bond lengths (Å) | 0.014 |
| Bond angles (°) | 1.5 |
| Avg. B factor (Å$^2$) | 9.46 |
| Wilson B factor (Å$^2$) | 7.2 |
| Ramachandran (%) | |
| Favored | 100% |
| Allowed | 0% |
| Outliers | 0 |

[*]Highest resolution shell shown in parenthesis.

DOI: https://doi.org/10.7554/eLife.46924.005

strand from residues 16–22 has also been observed in the full length in vitro fibrillar structure determined by cryoEM (*Figure 1—figure supplement 2*) (*Gremer et al., 2017*).

The antiparallel architecture and lack of registration of Aβ16–26 suggest this crystalline 'fibrillar'-like assembly has some characteristics of an amyloid oligomer. Structural studies of amyloid oligomers most frequently reveal anti-parallel β sheet architecture (*Tay et al., 2013*; *Laganowsky et al., 2012*; *Sarkar et al., 2014*), whereas fibril structures have revealed parallel β sheets (*Lührs et al., 2005*; *Colvin et al., 2016*; *Wälti et al., 2016*; *Krotee et al., 2018*), with the exception of some short segments of Aβ (*Colletier et al., 2011*) and in Aβ$_{1-40}$ containing the early onset hereditary mutation

D23N which leads to in-register anti-parallel fiber deposition in plaques (*Qiang et al., 2012*; *Tycko et al., 2009*). The out-of-register stacking of anti-parallel β strands has been proposed to be the defining trait of toxic oligomers (*Laganowsky et al., 2012*; *Liu et al., 2012*). The segment Aβ16–22 has been proposed to be able to form such oligomers in silica (*Sun et al., 2018*). The structures of Aβ16–21, Aβ16–26, and the full length fibrils may offer clues to designing inhibitors that impede both fibrillar and oligomeric assemblies.

## Efficacy of inhibitors of Aβ aggregation designed against Aβ 16–26 D23N

As the zipper motif observed in the atomic structure of $Aβ_{16-26}$ D23N may be relevant to a variety of amyloid beta assemblies, we sought to use it to develop structure-based peptide inhibitors of $Aβ_{1-42}$. Our laboratory has developed a Rosetta-based design strategy using steric zipper structures to design capping peptide inhibitors for a number of amyloid proteins implicated in disease (*Sievers et al., 2011*; *Seidler et al., 2018*; *Saelices et al., 2015*; *Soragni et al., 2016*; *Krotee et al., 2018*). We chose to truncate our structure to residues 16–22 for the search model, omitting the residues not in the β strand. We threaded amino acids onto a capping β strand and minimized energies of sidechains. From our first round of design, we chose six distinct inhibitor candidates; those that were identified as good candidates but containing strong amino acid similarities to other top inhibitors were discarded. Our initial pool of inhibitors contained four L-form peptides, 2 each of 6 and 8 amino acids length, termed L1-L4, and two D-peptides six amino acids long, termed D1 and D2.

We assessed the efficacy of the inhibitors at a 10 molar excess by testing if they prevented $Aβ_{1-42}$ toxicity on Neuro-2a (N2a) cells, a mouse neuroblastoma cell line, (*Olmsted et al., 1970*). We measured cytotoxicity using 3-(4,5-dimethylthiazol-2-yl)−2,5-diphenyltetrazolium bromide (MTT) dye reduction (*Mosmann, 1983*; *Liu et al., 1997*). Our toxicity assay revealed one inhibitor, D1 with the sequence (D)-LYIWVQ, that was able to eliminate the toxic effect of $Aβ_{1-42}$ (*Figure 2A*, *Supplementary file 1*); none of the inhibitors were toxic to N2a cells alone (*Figure 2—figure supplement 1A*). In our molecular model of the inhibitor, smaller hydrophobic residues of D1 mimic interactions with the fibril interface on one side of the peptide, which promotes recognition, (*Figure 2B*), while the other side of the peptide positions large aromatic residues between Aβ residues, blocking possible further interactions (*Figure 2C*).

We focused on these key features of the inhibitor sequence for our second round of design and aimed to improve efficacy. We lengthened our peptides to extend over more of our available structure towards the carboxy-terminus and made conservative residue changes to the face containing smaller hydrophobic residues. We selected and tested six new designs. Of the six, four were eight amino acids long such that the inhibitor would extend over more of our crystal structure, which we called D1a-D1d. The additional two, termed D1e and D1f, were six amino acids long featuring slight sequence perturbations from D1 (*Supplementary file 1*). We identified two of the eight amino acid long inhibitors, D1b and D1d, that were also effective at reducing $AB_{1-42}$ toxicity at both a tenfold excess and at an equimolar ratio (*Figure 2—figure supplement 1B,C*). We then tested these two inhibitors, as well as D1, across a range of concentrations with final concentrations ranging from 100 nM to 10 μM (*Figure 2D,E*). We found that all inhibitors elicited a dose dependent response, with all having an estimated $IC_{50}$ of less than 1 μM. The six residue long inhibitors, D1e and D1g, also had a similar effect on toxicity reduction as D1, however they did not perform as well as D1 in additional characterization and were not explored further (*Figure 3—figure supplement 1B*). The cognate negative peptide control, LC, the L-form peptide of inhibitor D1, did not reduce toxicity (*Figure 2E*).

## Reduction of toxicity by designed inhibitors is explained by a reduction of $Aβ_{1-42}$ aggregation

We next sought to understand the mechanism by which our peptide inhibitors reduce the toxic effect of $Aβ_{1-42}$. We therefore assayed fibril formation to discern if this reduction of toxicity could be explained by reduced aggregation. We incubated $Aβ_{1-42}$ with our inhibitors at 10:1, 1:1, and 1:10 molar ratios and monitored fibril formation by thioflavin-T (ThT) fluorescence at 37°C under quiescent conditions. We observe that all of our inhibitors reduce fibril formation in a dose dependent manner, while the negative control peptide, LC, does not (*Figure 3A*, *Figure 3—figure supplement 1A*).

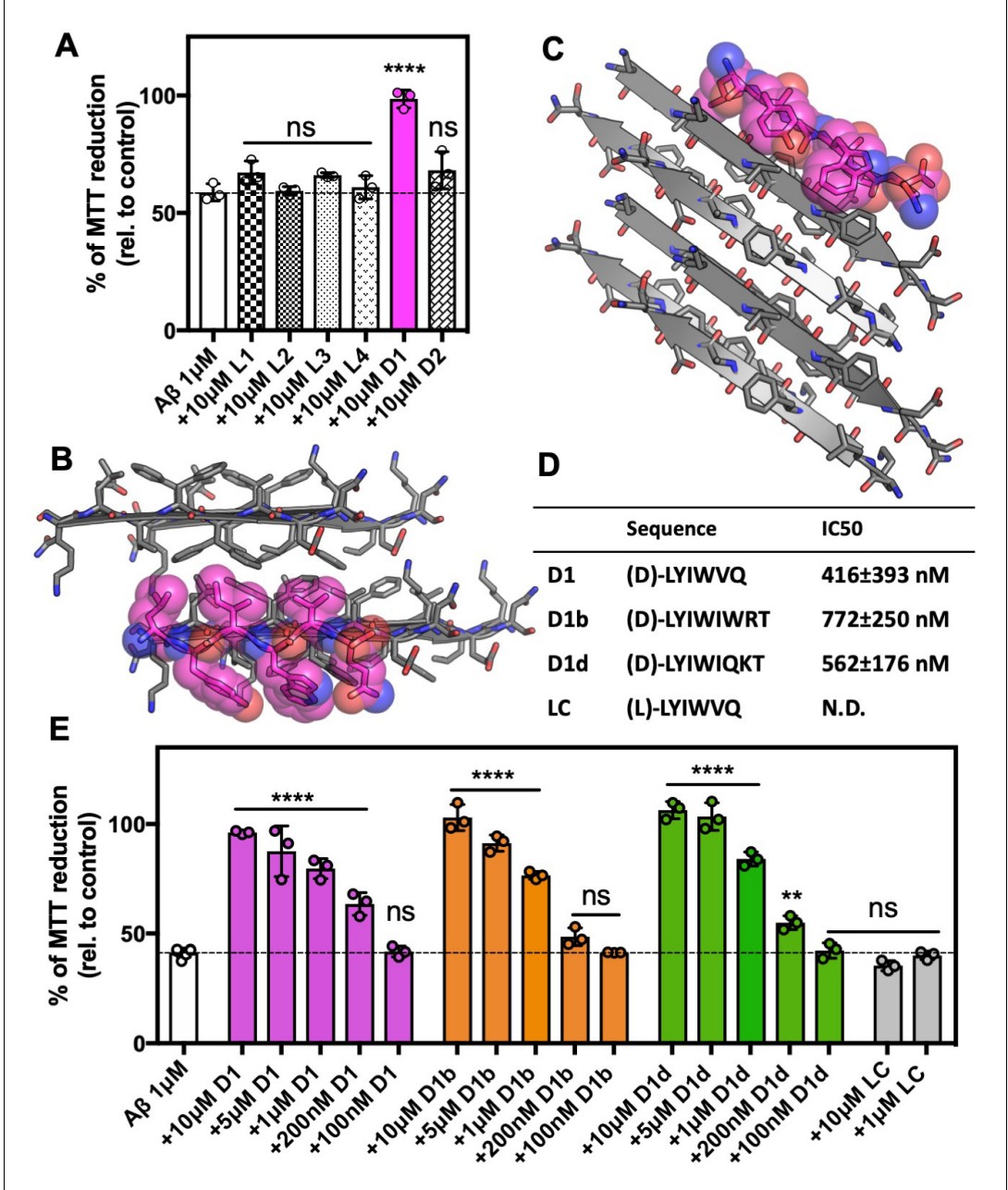

**Figure 2.** Development of inhibitors of Aβ fibril formation using structure-based design against Aβ 16–26 D23N. (A) Identification of Aβ1–42 inhibitor. 10 µM Aβ1–42 was incubated alone or with 100 µM of each candidate peptide inhibitor for 12 hr at 37°C and then diluted 1:10 with pre-plated N2a cells. Cytotoxicity was quantified using MTT dye reduction Bars represent mean with individual technical replicates, error bars display one standard deviation ($n = 3$; ns = not significant; ****, $p<0.0001$ using an ordinary one-way ANOVA- Dunnett's relative to leftmost column) (B, C) Segment KLVFFAEN, derived from the Aβ 16–26 D23N crystal structure, was used as the design target. Model of peptide inhibitor D1(magenta) bound to the design target, KLVFFAEN (gray). Smaller hydrophobic residues of D1 mimic interactions with the fibril interface on one side of the peptide (B), whereas the other side of the peptide positions large aromatic residues between Aβ residues, breaking possible further interactions (C). (D) Overview of peptide inhibitors in D and L amino acid conformations, as indicated, used in this study and their sequences. Peptide LC is the L-form cognate peptide of D-form peptide D1 and is the negative control for peptide inhibitor D1 and its derivatives D1b and D1d. $IC_{50}$ values were determined using four parameter nonlinear fit for half maximal inhibition. N.D., not determined. (E) Peptide inhibitors D1, D1b, and D1d reduce the cytotoxicity of Aβ1–42 in a dose dependent manner, whereas control peptide LC does not. 10 µM Aβ1–42 was incubated alone or with various concentrations of each peptide inhibitor for 12 hr at 37°C and then diluted 1:10 with pre-plated N2a cells. Cytotoxicity was quantified using MTT dye reduction. Bars represent mean with individual technical replicates, error bars display one standard deviation ($n = 3–6$; ns = not significant; **, $p<0.002$; ****, $p<0.0001$ using an ordinary one-way ANOVA Dunnett's relative to leftmost column).

DOI: https://doi.org/10.7554/eLife.46924.006

*Figure 2 continued on next page*

*Figure 2 continued*

The following source data and figure supplement are available for figure 2:

**Source data 1.** Toxicity data points—Aβ incubated with inhibitors and inhibitor controls.

DOI: https://doi.org/10.7554/eLife.46924.008

**Figure supplement 1.** Extended Toxicity data.

DOI: https://doi.org/10.7554/eLife.46924.007

The longer inhibitors, D1b and D1d, appear effective at an equimolar ratio. However, when assayed at higher concentrations, the inhibitors appear to self-assemble, but remain effective at reducing $Aβ_{1-42}$ toxicity (*Figure 3—figure supplement 1C*, *Figure 2E*). After 72 hr, samples were taken for negative-stain TEM analysis, which confirmed the reduced abundance of Aβ1–42 fibrils. D1b and D1d were more effective at reducing fibril formation than D1, although all three inhibitors showed near equal efficiency in reducing toxicity. Fibrils were observed in the equimolar ratio sample of $Aβ_{1-42}$ with D1, whereas the comparable samples with D1b and D1d did not contain fibrils. Inhibitors that were not efficient at preventing toxicity, such as D1a and D1c, were also less effective at blocking fiber formation (*Figure 2—figure supplement 1C*, *Figure 3—figure supplement 1A*).

Since oligomers, and not fibrils, are considered to be the more toxic species of Aβ (*Lesné et al., 2006*; *Lambert et al., 1998*; *Benilova et al., 2012*; *Jin et al., 2011*), we then investigated if our inhibitors affect the formation of oligomers or other cytotoxic $Aβ_{1-42}$ species. We used conformational antibodies to probe samples of $Aβ_{1-42}$ incubated with a 10-molar excess of inhibitor overnight at 37°C. Binding by oligomer specific conformational antibody A11 and A11-O9, a monoclonal variant of A11, was reduced by all of our inhibitors (*Figure 3C*, *Figure 3—figure supplement 1C*, *Figure 3—figure supplement 2*). While we have not determined the exact oligomeric assemblies the inhibitors are reducing, our antibody binding data coupled with the results of our toxicity assays suggest that the formation of a toxic oligomeric assembly is decreased. Additionally, the inhibitors reduced the abundance of Aβ conformations recognized by antibodies mOC24, mOC64, mOC104, and mOC116. These antibodies bind fibrillar plaques from patient derived AD tissue and/or 3xTg-AD mouse tissue (*Hatami et al., 2014*). Overall, these results indicate that our inhibitors may reduce oligomers, as well as disease relevant fibrillar conformations.

## Inhibitors bind and reduce toxicity of Aβ aggregates

As AD is only diagnosable long after Aβ aggregation has initiated, we wondered if these inhibitors would not only prevent amyloid aggregates from forming, but also if they can reduce the toxic effect of already formed aggregates. First, we incubated 10 μM Aβ at 37°C for 12 hr to form oligomers (*Figure 4—figure supplement 1A*), and then added inhibitors at various concentrations just prior to addition to N2a cells and assayed toxicity by MTT dye reduction. We found that adding the inhibitor to monomeric $Aβ_{1-42}$ prior to incubation had a marked difference from adding inhibitor to preformed $Aβ_{1-42}$ oligomers. When co-incubated with monomeric Aβ, the shorter D1 inhibitor was as effective as D1b and D1d at reducing toxicity; however, when added to pre-formed Aβ assemblies, only the longer inhibitors D1b and D1d were effective at reducing toxicity (*Figure 4A*). Both of the longer inhibitors could fully ameliorate toxicity of aggregates at 10 μM, but D1d is more potent, with effective reduction of toxicity to 1 μM. D1b differs from D1d only at amino acid positions 6 and 7. We suspect the difference in efficacy is conferred from residue 6, because both inhibitors contain positively charged residues at position 7, but at position 6 D1b contains a Gln while D1d has a much bulkier Trp. Our results indicate that while peptide inhibitors can both prevent aggregation initiation and block toxicity of aggregated assemblies, the latter appears to be more sensitive to slight perturbations in inhibitor composition.

We next performed TEM to determine if our inhibitors could disaggregate fibers, or if the fibers are being capped, as our inhibitor design would predict. We aggregated 10 μM $Aβ_{1-42}$ for 72 hr at 37°C under shaking conditions, then added inhibitors at 100 μM and incubated overnight. As the fibers are still present, we presume that our inhibitors are indeed capping or coating the fibers at toxicity inducing interfaces, thus preventing further seeding or toxic effects (*Figure 4B*). To investigate the capping ability of our inhibitors, we added the inhibitors to $Aβ_{1-42}$ during the exponential phase of fibril growth (*Figure 4—figure supplement 1B*). We found that even at the lowest

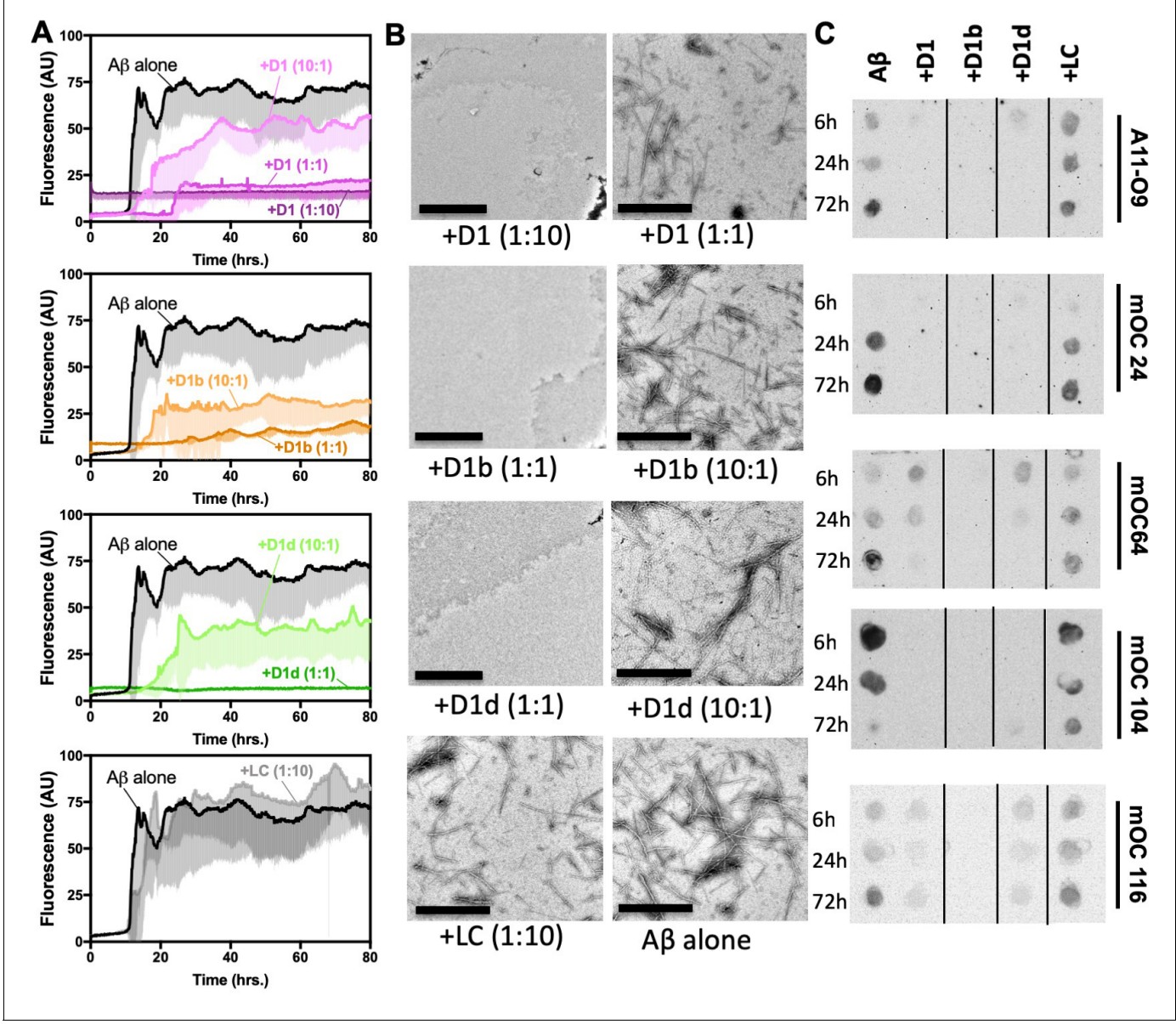

**Figure 3.** Designed inhibitors reduce aggregation of Aβ1–42. (**A**) Peptide inhibitors D1, D1b, and D1d reduce fibril formation of Aβ1–42, while negative control peptide LC does not. 10 μM of Aβ1–42 was incubated alone or at a 1:10, 1:1, or 10:1 molar ratio to each inhibitor under quiescent conditions at 37°C. Fibril formation was monitored using ThT fluorescence. Curves show the average of three technical replicates with one standard deviation below. (**B**) Negative-stain TEM analysis confirms the results of the ThT assays in *Figure 3A*. Samples were prepared as above and incubated for 72 hr before TEM analysis. Images of Aβ1–42 to D1 (1:10), D1b (1:1) and D1d (1:1) were captured at 3200x; scale bars are 2 μm. All other images were captured at 24,000x; scale bars are 500 nm. (**C**) Peptide inhibitors reduce the formation of Aβ1–42 assemblies recognized by conformational monoclonal antibodies, while negative control peptides do not. 10 μM Aβ1–42 was incubated alone (left-most column) or with 10-fold molar excess of each peptide-based inhibitor. Aliquots of the reaction were tested for antibody-binding at 6 hr, 24 hr, and 72 hr. Membranes were spliced as indicated for clarity.

DOI: https://doi.org/10.7554/eLife.46924.009

The following source data and figure supplements are available for figure 3:

**Source data 1.** Dot blot quatification data points.
DOI: https://doi.org/10.7554/eLife.46924.012
**Figure supplement 1.** A Peptide inhibitors D1, D1b, and D1d reduce fibril formation of Aβ1–42, while negative control peptide LC does not.
DOI: https://doi.org/10.7554/eLife.46924.010
**Figure supplement 2.** Extended dot blot data.
DOI: https://doi.org/10.7554/eLife.46924.011

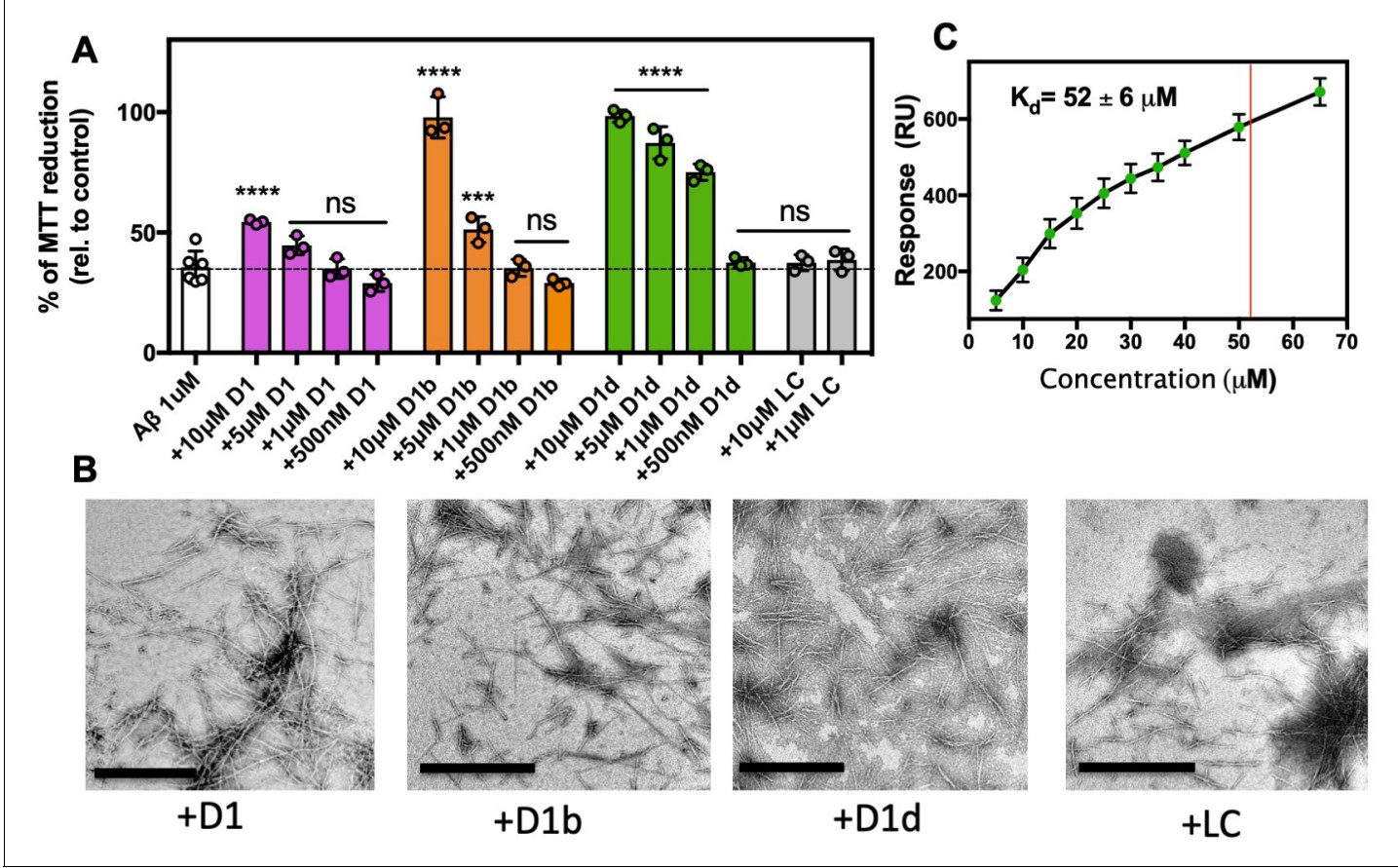

**Figure 4.** Inhibitors bind and block toxicity of aggregated Aβ1–42. (**A**) The toxicity of already formed Aβ1–42 aggregates is lessened by peptide inhibitors. 10 μM Aβ1–42 was incubated alone for 12 hr at 37˚C. Indicated molar ratio of inhibitor was added to the incubated Aβ1–42 and then diluted 1:10 with pre-plated N2a cells. Cytotoxicity was quantified using MTT dye reduction. Bars represent mean with individual technical replicates ($n$ = 3–6; ns = not significant; ***, $p<0.0005$; ****, $p<0.0001$ using an ordinary one-way ANOVA- Dunnett's relative to leftmost column). (**B, C**) Inhibitors bind to Aβ1–42 fibrils. (**B**) Peptide inhibitors do not disaggregate Aβ. 10 μM Aβ1–42 was incubated alone for 72 hr at 37 ˚C. Peptide inhibitors were added at 10-fold molar excess and incubated at RT for 24 hr before TEM analysis. Images were captured at 24,000x; scale bars are 500 nm. (**C**) Binding isotherm of inhibitor D1d to fibrillar Aβ1–42. The maximal response ($RU_{max}$) was derived by fitting sensorgrams obtained over a range of D1d concentrations to the binding model with a $K_d$ of 52 ± 6 μM, displayed as a red line. These $RU_{max}$ values are plotted (mean ± SD, n = 3) as a function of concentration and fitted to a one-to-one binding model, displayed as a black line.

DOI: https://doi.org/10.7554/eLife.46924.013

The following source data and figure supplements are available for figure 4:

**Source data 1.** Toxicity data points—Aβ aggregates treated with inhibitors.
DOI: https://doi.org/10.7554/eLife.46924.016

**Figure supplement 1.** Aggregated Aβ conformations and inhibitor capping of Aβ fibrils.
DOI: https://doi.org/10.7554/eLife.46924.014

**Figure supplement 2.** Representative Sensorgram obtained when D1d solutions at the indicated concentrations were flowed across the Aβ1–42 sensor chip.
DOI: https://doi.org/10.7554/eLife.46924.015

concentration of inhibitor, 10 μM Aβ: 1 μM inhibitor, we see minimal increase of signal for inhibitors D1b and D1d. Additionally, we observed a slight lowering of ThT signal samples with a 1:1 inhibitor addition, possibly due to inhibitors displacing ThT molecules bound to the fibrils. As the inhibitors do prevent monomer aggregation as well (**Figure 3A**), we are cautious to overinterpret the result of this experiment, as the inhibitor could feasibly be sequestering free monomer or small assemblies from adding to the fibrils.

We performed SPR to verify that our inhibitors bind to fibers. We find that the most potent inhibitor of aggregated assemblies, D1d, binds to Aβ1–42 fibrils with an apparent $K_d$ of 52 μM (**Figure 4C**,

*Figure 4—figure supplement 2*). We used a one-inhibitor-to-one-protein substrate model to fit the data; however, the true $K_d$ may be lower due to the complication of D1d self-interaction and polymorphic Aβ fibrils. Thus, we have shown that inhibitors D1b and D1d not only prevent aggregation of monomeric Aβ, but also bind aggregated states.

## Inhibitors reduce seeding of tau by aggregated Aβ$_{1-42}$

Having demonstrated that our inhibitors block a toxic interface on Aβ, we next questioned if this interface could also be involved in cross seeding tau. First, we sought to validate the direct seeding mechanism that has been reported by others (*Guo et al., 2006*; *Miller et al., 2011*; *Vasconcelos et al., 2016*). We tested seeding of the microtubule binding domain of tau, K18 + (244-380) in a ThT assay at 37°C under shaking conditions and found that fibrils of Aβ$_{1-42}$ and Aβ$_{16-26}$ D23N seeded aggregation, though not as efficiently as fibrils of K18 (*Figure 5A*, *Figure 5—figure supplement 1A*). This seeding effect was also observed on full length tau in the presence of heparin (*Figure 5—figure supplement 1B*). Conversely, K18 was unable to seed Aβ (*Figure 5—figure supplement 1C*).

Next, we tested seeding in a well-established HEK293 biosensor cell line, tau-K18 (P301S) EYFP, which stably expresses the microtubule binding domain of tau P301S mutant. This cell line, referred to hereafter as tau-K18 biosensor cells, has been used to demonstrate prion like seeding from transfected tau fibrils to cells and has been used as a model system to test tau inhibitors (*Seidler et al., 2018*; *Kfoury et al., 2012*). We transfected biosensor cells with tau40 or Aβ fibrils (*Figure 4—figure supplement 1A*) to a final concentration of 250 nM. We found that Aβ was able to produce intracellular aggregates significantly greater than the vehicle alone, but only at around 2.5% efficiency of tau40. It is not altogether surprising that Aβ has such a low efficiency of cross-seeding; this mirrors a previous result in a similar system (*Vasconcelos et al., 2016*). It is possible that tau fibrils contain multiple polymorphs and interfaces capable of homotypic seeding, whereas Aβ may have a more limited number of tau seeding-competent conformations. Additionally, in vitro Aβ aggregation may create disproportionate ratios of assemblies compared to those present in AD. Regardless, it remains that some Aβ species is tau-seeding competent. The finding that Aβ is indeed able to seed aggregation in tau-K18 expressing cell lines suggests that the cross-interacting region of tau is located on this microtubule binding domain. We found other amyloid protein fibrils and non-fibrillar Aβ are not seeding-competent in this system (*Figure 5—figure supplement 1D*), indicating the biosensor cell assay can faithfully differentiate between fibrils of amyloid proteins, which differ in their underlying structures and sequences.

If our inhibitors block the interface responsible for seeding, we would expect Aβ treated with inhibitors to no longer to be seeds for tau. To test this hypothesis, we treated 250 nM Aβ fibers with indicated concentrations of inhibitor for 1 hr and transfected these into the biosensor cell line. All of our inhibitors were able to reduce seeding at 20 μM final concentration, while D1b showed a reduction in seeding at a concentration as low as1 μM (*Figure 5DE*). While both D1b and D1d reduced Aβ aggregate toxicity on N2a cells, D1d was the more effective inhibitor of Aβ toxicity, whereas D1b is the more effective inhibitor at reducing tau seeding.

We next sought to verify that the region of Aβ used to design inhibitors is important in seeding tau, and could be targeted by inhibitor D1b. We created two mutants of Aβ$_{1-42}$, with residues on either size of our steric zipper interface disrupted: Aβ$_{1-42}$ L17R/F19R and Aβ$_{1-42}$ K16A/V18A/E22A. We chose not to mutate residue Phe20, as it has been observed on both buried and solvent accessible interface in full length structures (*Lührs et al., 2005*; *Colvin et al., 2016*; *Gremer et al., 2017*). We formed fibrils of each mutant construct and wild type, then incubated these fibrils with the indicated concentration of D1b, and used this to seed tau-K18 biosensor cells, as described previously (*Figure 5—figure supplement 2*). We found that the fibrils of Aβ$_{1-42}$ L17R/F19R were able to seed similarly to WT Aβ$_{1-42}$, while no seeding was detected from Aβ$_{1-42}$ K16A/V18A/E22A. Seeding by Aβ$_{1-42}$ L17R/F19R was inhibited by D1b, suggesting that residues K16, V18 and E22 create the seeding interface which is targeted by inhibitor D1b.

## Inhibitors reduce tau aggregation and seeding

Our data support previous studies that suggest the tau binding surface on Aβ is localized to the segment whose structure we determined and targeted for design of inhibitors against Aβ aggregation

A

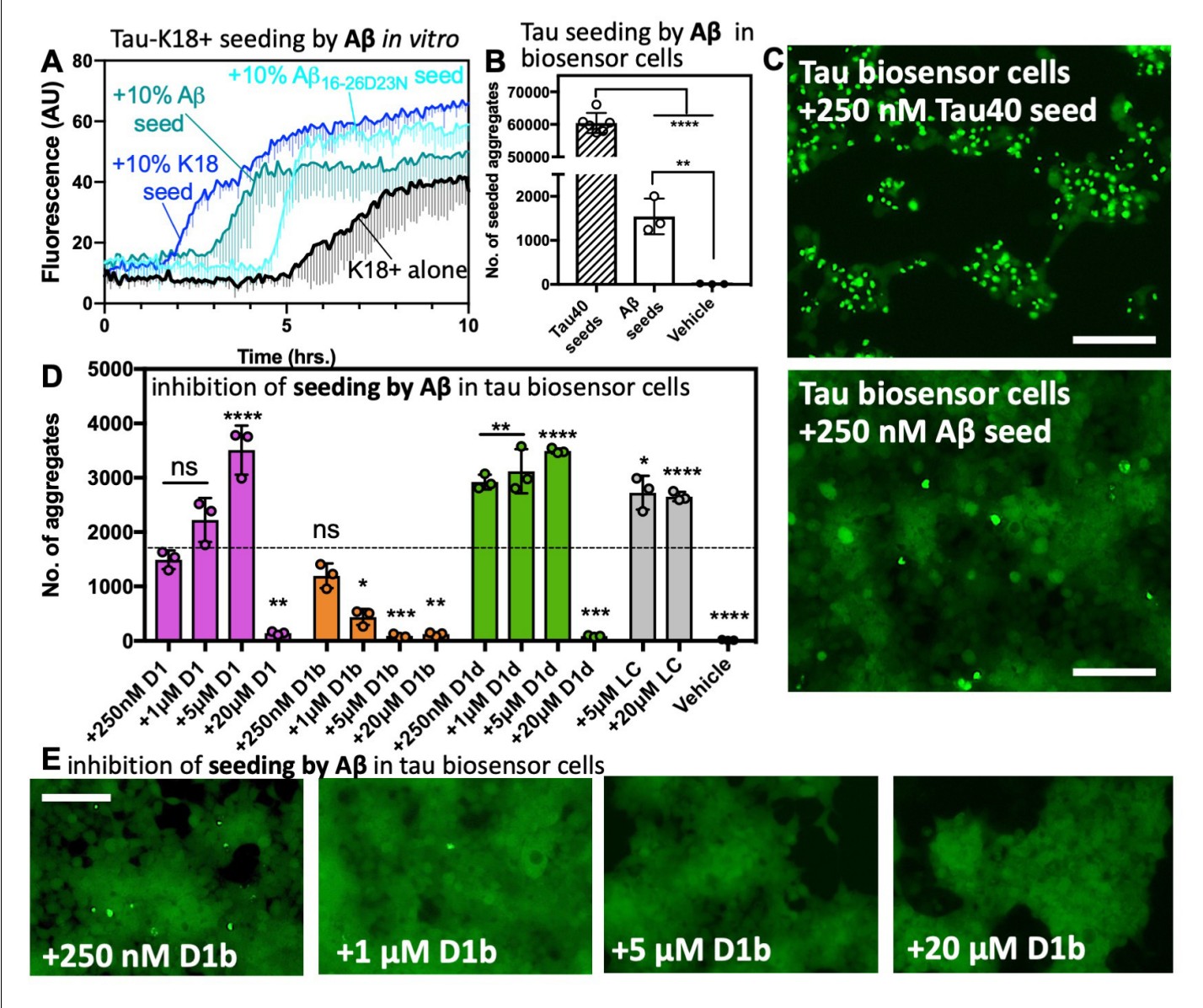

**Figure 5.** Tau aggregation is seeded by Aβ and reduced by structure-based inhibitors. (A) 50 µM tau-K18+ was seeded with 10% monomer equivalent of pre-formed fibrils of Aβ1–42, Aβ16-26 D23N or tau-K18 under shaking conditions at 700 RPM at 37°C in PBS. Fibril formation was monitored using ThT fluorescence. Error bars below show the standard deviation of the average of three technical replicates. (B) The number of intracellular aggregates present in tau-K18CY biosensor cells normalized to cell confluence seeded by the addition of 250 nM tau40 or 250 nM Aβ1–42 fibrils. Error bars show the standard deviation of the mean of technical replicates (n = 3; ****, p=0.0001 using an ordinary one-way ANOVA- Dunnett's relative to leftmost column, and **, p=0.0028 in unpaired t test of Aβ vs. vehicle) (C) Representative images of seeded cells from B at 10x magnification, scale bar 100 µm. (D and E). Concentration dependent inhibition of Aβ1–42 induced seeding of tau aggregation in tau-K18CY biosensor cells. (D) Average seeding by Aβ as a function of indicated inhibitor concentration. Error bars show the standard deviation of the mean of technical replicates (n = 3; ns = not significant; *, p<0.02; **, p<0.005; ***, p<0.001; ****, p<0.0001 using an ordinary one-way ANOVA- Dunnett's relative to leftmost column), and the dotted line shows the mean number of aggregates from untreated Aβ1–42 fibrils. (E) Representative images of tau-K18CY biosensor cells showing the concentration-dependent effect of D1b on seeding. Cells are shown at 10X magnification, scale bar 100 µm.

DOI: https://doi.org/10.7554/eLife.46924.017

The following source data and figure supplements are available for figure 5:

**Source data 1.** Tau biosensor seeding data points—Aβ and other amyloid fibrils with and without inhibtors.
DOI: https://doi.org/10.7554/eLife.46924.020

**Figure supplement 1.** Extended ThT and seeding data.
DOI: https://doi.org/10.7554/eLife.46924.018

*Figure 5 continued on next page*

*Figure 5 continued*

**Figure supplement 2.** Seeding and inhibition of Aβ interface mutants.

DOI: https://doi.org/10.7554/eLife.46924.019

and Aβ-mediated seeding of tau (*Guo et al., 2006*; *Miller et al., 2011*). We hypothesized that the tau fibril could contain a similar self-complementary surface and would also be susceptible to treatment with our inhibitors. We first asked if the Aβ inhibitors, D1, D1b, and D1d could prevent monomeric tau from aggregating. We performed a ThT assay on 10 μM tau40, at 37°C with shaking and 0.5 mg/mL heparin and found that all inhibitors function in a dose dependent manner similar to our results with Aβ monomer, while the control inhibitor LC does not reduce tau aggregation (*Figure 6A*, *Figure 6—figure supplement 1A*). The peptide inhibitors are not able to block aggregation of the amyloid forming proteins hIAPP or alpha synuclein, indicating that these inhibitors are specific for Abeta and tau, and are not general amyloid inhibitors (*Figure 6—figure supplement 1B*).

Because we had observed differences in inhibitor efficacy on monomer versus aggregated species of Aβ, we next tested if the inhibitor was effective against the seeding ability of tau40 fibrils. We formed tau40 fibrils, treated them with indicated inhibitor concentration and transfected into tau-K18 biosensor cells to measure seeding inhibition. We found that similar to our Aβ-mediated tau biosensor seeding experiment, D1b was the best inhibitor, with an $IC_{50}$ of 4.5 μM. D1 was slightly effective, while D1d showed seeding reduction only when increased to 75 μM (*Figure 6B,C*). It could be that D1b plays a dual role to inhibit both Aβ and tau, and this combined effect could explain the drastically reduced seeding from Aβ fibrils in our prior experiment (*Figure 5D*).

Next, we sought to determine potential binding sites on tau for D1b. We postulated that regions know to be drivers of tau aggregation could share structural features with the Aβ core, and thus be inhibited by D1b. We designed mutants of tau40 that disrupt key interactions in steric zipper interfaces determined from crystal structures of VQIINK (*Seidler et al., 2018*) and VQIVYK (*Sawaya et al., 2007*), and cryoEM models of AD tau fibrils (*Fitzpatrick et al., 2017*). In total we tested six different constructs, each designed to block all but one aggregation interface of tau. The first three mutants were engineered to block the VQIVYK aggregation interfaces in addition to all but 1 of the three different known VQIINK interfaces. Mutant 1 (Q276W, L282R, I308P) leaves only interface A of VQIINK available for aggregation, mutant 2 (Q276W, I277M, I308P) leaves only interface B for aggregation, and mutant 3 (I277M, L282R, I308P) leaves only interface C accessible for aggregation. Constructs 4 and 5 were designed to test the effect of blocking VQIINK and all but 1 of the VQIVYK surfaces. Mutant 4 (Q276W, I277M, L282R, Q307W, V309W) leaves only the dry interface of VQIVYK available for aggregation and mutant 5 (Q276W, I277M, L282R, I308W) leaves only the solvent accessible surface for aggregation. In addition, we tested the effect of D1b on blocking seeding by 3R tau, which lacks the VQIINK aggregation segment and leaves the VQIVYK interface intact (*Figure 6D*, *Figure 6—figure supplement 2C–E*).

To test if specific interfaces are inhibited by D1b, fibrils were formed from all of the different mutants, and then each was incubated with the indicated concentration of D1b and used to seed wild type tau-K18 biosensor cells, as described previously with wild type tau fibrils (*Figure 6—figure supplement 3A,B*). We found that D1b was most effective at inhibiting seeding by fibrils of mutants that left intact: interface A of VQIINK which is thought to involve aggregation at site I277 of tau, the solvent accessible interface of VQIVYK as well as 3R tau (*Figure 6D*). D1b also showed moderate inhibition of several other tau mutants, but required high concentrations to inhibit seeding (*Figure 6—figure supplement 2C*). As a control, we tested seeding by a mutant of tau40 that combined all of the different mutations, and found this mutant did not induce any seeding in tau-K18 biosensor cells (*Figure 6—figure supplement 1E*), despite forming fibrils when incubated with heparin, indicating that at least one of the known interfaces is needed for seeding. Control inhibitor LC has little to no effect on seeding from any construct (*Figure 6—figure supplement 1F*). Taken together, these data show that both the VQIINK and VQIVYK aggregation segments of tau are inhibited by D1b, and suggest that each may share common structural features with the Aβ core that could allow for cross-seeding of tau by Aβ.

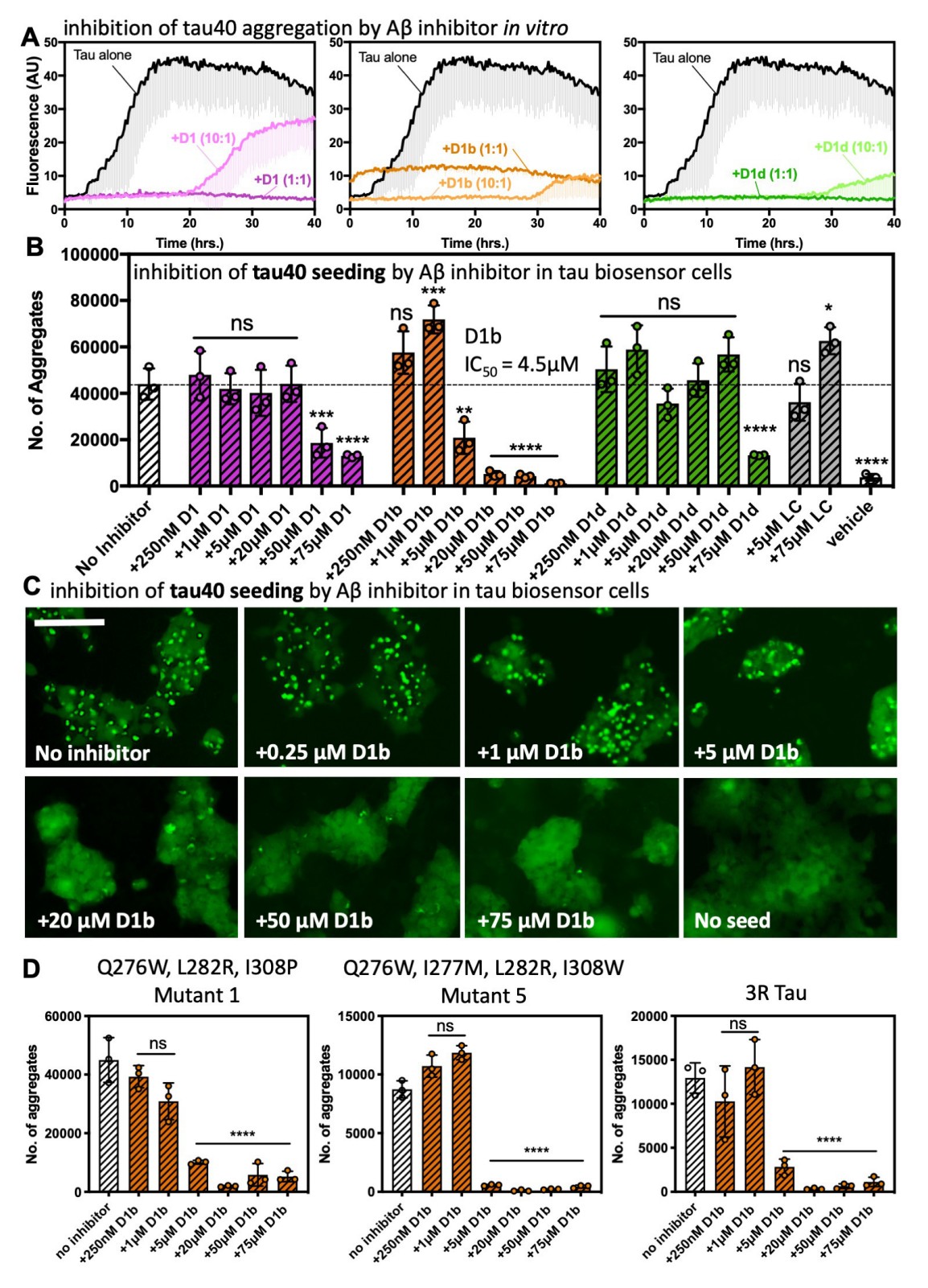

**Figure 6.** Aβ inhibitors also reduce fibril formation and seeding by tau40. (**A**) Peptide inhibitors D1, D1b, and D1d reduce fibril formation of tau40. 10 μM tau40 monomer was incubated at a 1:10, 1:1, or 10:1 molar ratio to each inhibitor with 0.5 mg/ml heparin under shaking conditions at 700 RPM at 37˚C. Fibril formation was monitored using ThT fluorescence. Plots show the average of three technical replicates with one standard deviation below. (**B**), (**C**). The effects of the inhibitors on seeding by tau40 fibrils in tau-K18CY biosensor cells. The cells were seeded with 250 nM tau40 fiber (final

*Figure 6 continued on next page*

*Figure 6 continued*

concentration); in samples with inhibitor, tau40 fibers were incubated with indicated final concentrations of peptide inhibitor for one hour prior to addition to cells. (B) Average number of aggregates at the indicated inhibitor concentrations, Bars represent mean with individual technical replicates, error bars display one standard deviation ($n = 3$; ns = not significant; *, $p<0.03$; **, $p<0.023$; ***, $p<0.0008$; ****, $p<0.0001$ using an ordinary one-way ANOVA- Dunnett's relative to leftmost column). dotted line represents number of aggregates from untreated tau40 fibrils. $IC_{50}$ value was calculated from the dose–response plot of inhibitor D1b. (C). Representative images of effect of D1b on seeding. Cells are shown at 10X magnification, scale bar 100 μm. (D) Seeding from tau interface mutation fibrils in tau-K18CY biosensor cells is reduced by D1b. Experiment was performed as above. Average number of aggregates at the indicated inhibitor concentrations, Bars represent mean with individual technical replicates, error bars display one standard deviation ($n = 3$; ns = not significant; ****, $p<0.0001$ using an ordinary one-way ANOVA- Dunnett's relative to leftmost column).
DOI: https://doi.org/10.7554/eLife.46924.021

The following source data and figure supplements are available for figure 6:

**Source data 1.** Tau biosensor seeding data points—tau and tau interface mutations with and without inhibitor.
DOI: https://doi.org/10.7554/eLife.46924.026
**Figure supplement 1.** Specificy of inhibitors.
DOI: https://doi.org/10.7554/eLife.46924.022
**Figure supplement 2.** Extended tau40 interface mutation data.
DOI: https://doi.org/10.7554/eLife.46924.023
**Figure supplement 3.** The spines of Aβ 16–26 D23N and tau are structurally similar.
DOI: https://doi.org/10.7554/eLife.46924.024
**Figure supplement 4.** Hetero-seeding model from side of Aβ fibril.
DOI: https://doi.org/10.7554/eLife.46924.025

## Designed inhibitor D1b targets disease relevant conformations

Amyloid polymorphs may differ depending on whether they were aggregated in vitro or extracted from human brain tissue (*Falcon et al., 2018*). We sought to determine if our inhibitors are capable of blocking pathological forms of either tau, or Aβ. As suggested previously in our conformational antibody assay and structural alignment (*Figure 3C*), we hypothesized that our inhibitors would block disease-relevant amyloid polymorphs. Since we also found that our inhibitors blocked both homotypic and heterotypic tau seeding by aggregated tau and Aβ, we tested our inhibitor series on crude lysate from AD donor patient brain tissue.

We homogenized tissue from three different brain regions of a single AD patient brain, the hippocampal region, affected early as classified by Braak staging, and frontal and occipital lobe regions, which are affected later in disease progression (*Brier et al., 2016*; *Schwarz et al., 2016*). We also prepared samples from patient tissue with progressive supranuclear palsy (PSP), which is a tau aggregation disease that displayed no Aβ aggregation by immunostaining. We prepared samples from tissue of a non-diseased patient, as well as tau-immunodepleting PSP tissue. We transfected brain lysates into the biosensor cells; samples with inhibitor were treated with 10 μM D1, D1b, or D1d.

We found that treating the brain-derived lysates with D1b significantly reduced seeding by all tested brain tissue samples (*Figure 7*). The tau aggregate load from the different tissues has not been controlled, and this is likely the reason for different seeding efficiencies that are observed from different tissue types. Although our inhibitor D1b showed reduction of seeding in the hippocampal sample, the fibril load of this region may have been too great to have been efficiently halted by the dose used. Interestingly, the PSP tauopathy tissue was also responsive to treatment with each of the inhibitors, with D1b displaying the most pronounced inhibition. We surmise that D1b recognizes a common toxic epitope found in both Aβ, and in a variety of tau polymorphs.

## Discussion

The search for druggable targets in AD is muddied by the numerous proteins involved and incomplete understanding of whether or not the two histological protein hallmarks, Aβ and tau, interact directly with each other. On top of this, Aβ, the apparent initiator of the disease, aggregates into a wide variety of species, from soluble oligomers ranging from dimers to those that contain dozens of copies, to polymorphic fibril deposits. While there may be numerous toxic assemblies, targeting a

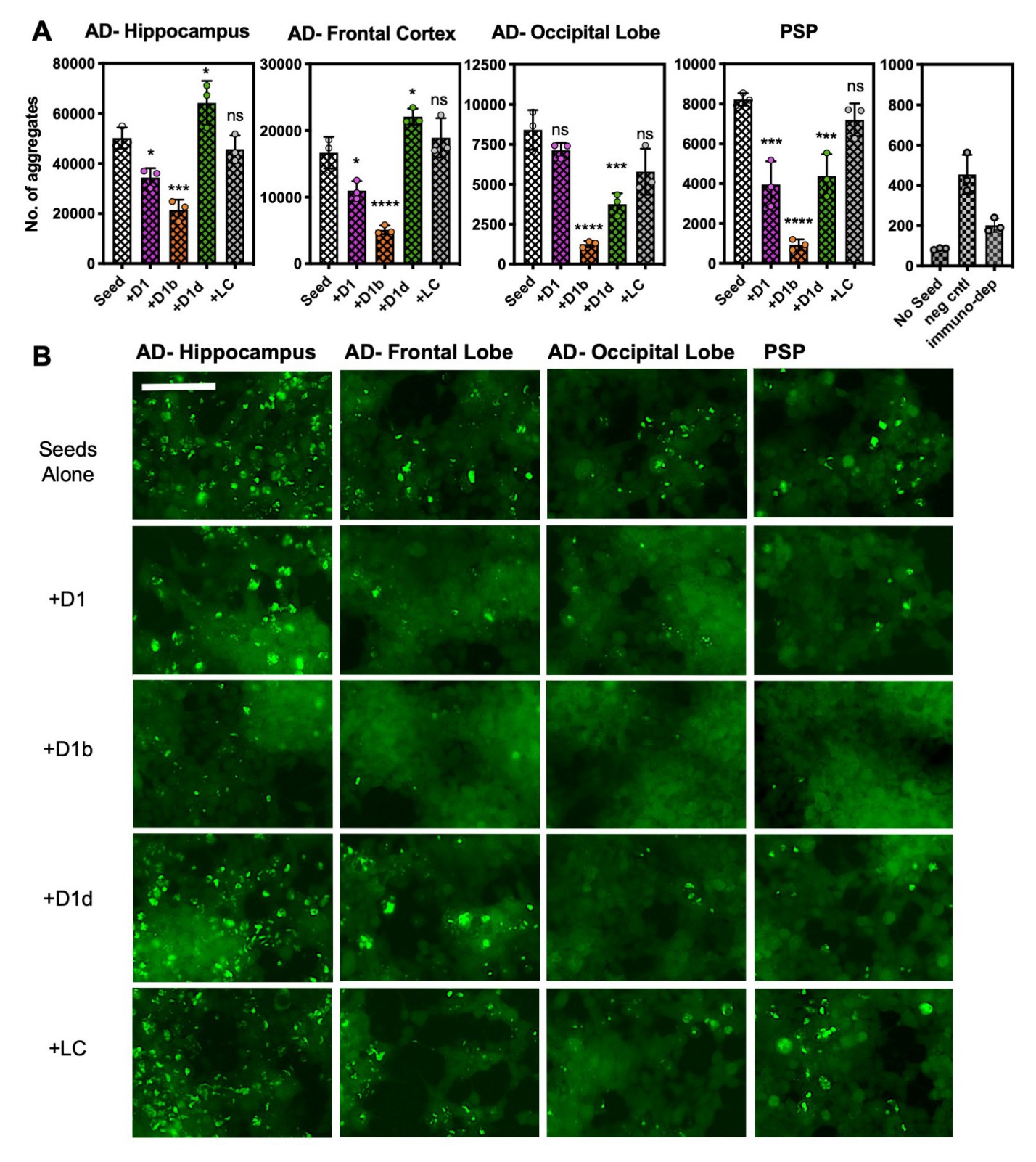

**Figure 7.** Peptide inhibitors reduce seeding by crude brain-extract from tauopathy donor tissue. Brain lysate was prepared in TBS buffer from three brain regions of one AD patient, and from a one sample of a PSP patient lacking Aβ plaques. Brain lysate from a non-disease patient (neg cntl) and a tau immunodepleted sample from PSP tissue are in right panel. Cells were seeded with a 1/400 dilution of brain tissue lysate; for samples with inhibitor, lysates were incubated with inhibitor overnight prior to addition to cells. A concentration of 10 µM peptide was used for all of the experiments

*Figure 7 continued on next page*

*Figure 7 continued*

shown. (**A**) The average number of aggregates seeded by lysate from each respective brain region, with or without addition of inhibitors. Bars represent mean with individual technical replicates (*n* = 3; ns = not significant; *, p<0.05; ***, p<0.0005; ****, p<0.0001 using an ordinary one-way ANOVA-Dunnett's relative to leftmost column). (**B**) Representative images of seeded biosensor cells from A shown at 10X magnification, scale bar 100 μm. Extended ANOVA data included as a supplementary file.

DOI: https://doi.org/10.7554/eLife.46924.027

The following source data is available for figure 7:

**Source data 1.** Tau biosensor seeding data points—brain lysate with and without inhibitor.

DOI: https://doi.org/10.7554/eLife.46924.028

specific sequence or structure of a toxic motif that is present in a variety of these assemblies could be an effective strategy for designing pharmaceuticals.

We targeted the amyloid core segment of Aβ due to its defined amyloidogenicity, and putative interaction with the late-stage aggregating protein, tau. We focused our efforts on the Aβ$_{16-26}$ segment with a hereditary mutation D23N, whose structure we determined by MicroED. Although the crystalline structure of this segment is fibril-like, and resembles a previously observed zipper interface as well as an interface in full length fibrils, the out-of-register interface of the β-strands suggests that portions of this conformation may be present in a number of toxic oligomeric intermediates as well as in fibrils. We successfully used this structure to design a series of related inhibitors that reduce toxicity of Aβ in model N2a cells.

Our biochemical and toxicity studies indicate that these inhibitors function in two ways. The first is by preventing monomeric Aβ from aggregating. The second is by reducing toxicity of pre-formed oligomeric Aβ, possibly by binding to and blocking a surface that is responsible for conferring toxicity or seeding. While all of our designed inhibitors prevent monomeric Aβ from aggregating, only the longer D1b and D1d versions are effective at reducing toxicity of preformed assemblies. These two peptides were designed by extending the C-terminus. D1b and D1d could conceivably act by obscuring resides important for conferring toxicity, as supported by early onset hereditary mutations clustering at residues 21–23 (*Lazo et al., 2009*; *Krone et al., 2008*; *Chong et al., 2013*).

Our data implicate an extended Aβ core in the spread of the disease, because targeting inhibitors to this region appears to block the templating interface needed to cross-seed tau. Aβ fibers treated with D1b showed a dramatic reduction of cross seeding in tau-K18 biosensor cells. Tau fibers treated with D1b showed similarly inhibited seeding in biosensor cells. The dual efficacy of the inhibitor D1b designed against the 16–23 region of Aβ suggests that these two pathological aggregates, Aβ and tau, share a common structural motif in AD. Indeed, we find that solvent accessible residues K16, V18, and E22 of Aβ are important for tau seeding. Conversely, by using mutant constructs of tau with only one available amyloid interface, we were able to determine two interfaces on tau where seeding was highly reduced by D1b. Both the R2 and R3 amyloid-prone regions of tau contain a D1b sensitive interface. Of note, the surface of R2 blocked by D1b contains residue I277, which previously has been shown to be critical for tau aggregation (*Kirschner et al., 1986*).

We find that overlaying our Aβ segment crystal structure with structures of tau R2 and R3 reveals a high degree of structural similarity both in the backbone, and also apparently in the complementarity of sidechains from each to intrinsically interdigitate (*Figure 6—figure supplement 3*). Interestingly, Aβ overlays well with these regions of tau in both parallel and anti-parallel orientations, suggesting that either fiber or smaller oligomers could be capable of cross seeding on the ends of fibrils. Another mode of seeding could be facilitated along the side of the Aβ fibril using the solvent accessible interface from residues 16–22 (*Figure 6—figure supplement 4*). While the modeled interface with tau is calculated to form with a favorable energy, burying of polar and charged residues (TauQ307/AβK16) could cause this interaction to be transient. Additionally, the differences in overall structure and stacking twists of the two fibrils could also explain why fibrils incorporating both proteins or fibril bundles containing both Aβ and tau fibrils are not observed. On this basis, we suggest that the amyloid core of Aβ and the regions VQIINK and VQIVYK can form similar structures in AD that are biochemically capable of cross-seeding.

Consistent with our finding that regions of Aβ and tau share structural similarities, we found that the Aβ inhibitor D1b is able to reduce seeding from brain homogenates, indicating that the inhibitor

is recognizing a disease-related structural motif, while D1 and D1d are much less effective. It is curious that seeding by both AD and PSP is greatly reduced by the inhibitor D1b, as PSP pathology does not include Aβ aggregates. It is thought that different disease phenotypes, which display distinct fiber morphologies commonly referred to as strains, are determined by the formation of different steric zipper cores (*Sanders et al., 2014*). Thus, PSP fibers may contain a different core than our in vitro aggregated tau or AD derived tau. However, our tau mutagenesis results suggest that inhibitor D1b can recognize at least two unique core interfaces, and thus could be able to act on multiple strains of tau fibers. There may exist other fibril polymorphs with different structural conformations which are not sensitive to D1b. Furthermore, it is unknown how different co-factors and post-translational modifications, such as tau phosphorylation, could affect the ability of Aβ to seed tau, and thus efficacy of D1b on tau seeding. Aβ cross-seeding may represent one of many possible stimuli of tau aggregation.

Similar to other peptide-based amyloid inhibitors, the effective dose to reduce toxicity of aggregated species is higher than to delay aggregation of monomeric species. This is emphasized by the differing efficacies of our related inhibitor series, where some inhibitors were able to prevent initial aggregation, but not toxicity or seeding from various assemblies. It appears that inhibitors to prevent an aggregation nucleus are much more promiscuous than those that ameliorate toxicity by binding to a distinct structure. This trend was observed in both Aβ and tau, suggesting a common inhibitory mechanism for both proteins, and highlights the need for multiple experimental measures to validate inhibitor efficacy.

In summary, our results suggest that a direct interaction between the Aβ core and the amyloid-prone regions of tau facilitates cross seeding. Our inhibitors designed for the Aβ core segment prevent cross seeding of tau, as well as tau homotypic seeding. The entwined nature of these two proteins in AD suggests it is necessary to control aggregation of both in order to treat the disease. Early detection is still crucial, but these data provide a platform on which further inhibitors can be designed for optimized inhibition of amyloid seeding in Alzheimer's disease.

# Materials and methods

## Key resources table

| Reagent type | Designation | Source or reference | Identifiers | Additional information |
|---|---|---|---|---|
| Cell line (*Homo-sapiens*) | HEK293 -K18 (P301S)-EYFP | Diamond Laboratory | | |
| Cell line (*M. musculus*) | Neuro-2a cell line | ATCC | Cat # CCL-131 RRID:CVCL_0470 | |
| Antibody | Goat anti-Mouse IgG H and L (FITC) secondary antibody | Abcam | Cat# ab7064, RRID:AB_955234 | WB 1:10000 |
| Antibody | Donkey anti-Rabbit IgG H and L (FITC) secondary antibody | Abcam | RRID:AB_955259 | WB 1:10000 |
| Antibody | A11 (rabbit polyclonal) | Millipore | Cat# AB9234 RRID:AB_11214948 | WB 1:500 |
| Antibody | OC (rabbit polyclonal) | Millipore | Cat# AB2286 RRID:AB_1977024 | WB 1:2000 |
| Antibody | 6E10 (mouse monoclonal) | BioLegend | Cat# 803003, RRID:AB_2564652 | WB 1:5000 |
| Antibody | A11-09 mOC 64 mOC 24 mOC 116 mOC 104 (rabbit monoclonal) | Glabe Laboratory, Abcam | | WB 1:100 |

*Continued on next page*

*Continued*

| Reagent type | Designation | Source or reference | Identifiers | Additional information |
|---|---|---|---|---|
| Peptide, recombinant protein | Recombinant Amyloid Beta 1–42 (pET15b-MBP-AB) | This paper | | See Materials and methods section |
| Peptide, recombinant protein | (D)-LYIWVQ | Genscript | | >95% purity |
| Peptide, recombinant protein | (D)-LYIWIWRT | Genscript | | >95% purity |
| Peptide, recombinant protein | (D)-LYIWIQKT | Genscript | | >95% purity |
| Peptide, recombinant protein | (L)-LYIWVQ | Genscript | | >95% purity |
| Peptide, recombinant protein | Recombinant Tau40 (1–441, pET22b) | This paper | | See Materials and methods section |
| Peptide, recombinant protein | Recombinant K18(244-372) K18+ (244-380) (PNG2) | This paper | | See Materials and methods section |
| Peptide, recombinant protein | KLVFFAENVGS | Genscript | | >98% purity |
| Chemical compound, drug | 3-(4,5-dimethylthiazol-2-yl)−2,5-diphenyltetrazolium bromide (MTT) dye | Sigma | Cat# M5655 | |
| Chemical compound, drug | ThioflavinT | Sigma | CAS ID: 2390-54-7 | |
| Software, algorithm | XDS | http://xds.mpimf-heidelberg.mpg.de/ | RRID:SCR_015652 | |
| Software, algorithm | CCP4 | http://www.ccp4.ac.uk/ | RRID:SCR_007255 | |
| Software, algorithm | PHENIX | PMID: 20124702 | http://www.phenix online.org/; RRID: SCR_014224 | Model building and refinement |
| Software, algorithm | Coot | PMID: 20383002 | https://www2.mrc-lmb.cam.ac.uk/personal/pemsley/coot/; RRID: SCR_014222 | Model building |
| Software, algorithm | Rosetta | https://www.rosettacommons.org/home | RRID:SCR_015701 | |
| Software, algorithm | Graphpad Prism | GraphPad Prism (https://graphpad.com) | RRID:SCR_015807 | Version 8 |
| Software, algorithm | ImageJ | ImageJ (http://imagej.nih.gov/ij/) | RRID:SCR_003070 | |
| Software, algorithm | Foldit | http://fold.it/ | RRID:SCR_003788 | |

## Recombinant Amyloid Beta Peptide purification

Aβ, and interface mutants, were purified as described in *Krotee et al. (2018)*. After purification, the protein was lyophilized. Dried peptide powders were stored in desiccant jars at −20°C.

## Peptide Preparation

Candidate inhibitors were custom made and purchased from Genscript (Piscataway, NJ). Lyophilized candidate inhibitors were dissolved at 10 mM in 100% DMSO. 10 mM stocks were diluted as necessary. All stocks were stored frozen at −20˚C.

Amyloid Beta was prepared by dissolving lyophilized peptide in 100% DMSO or 100 mM $NH_4OH$. Next, the sample was spin-filtered and the concentration was assessed by BCA assay (Thermo Scientific, Grand Island, NY). The DMSO or $NH_4OH$ peptide stocks were diluted 100-fold in filter-sterilized Dulbecco's PBS (Cat. # 14200–075, Life Technologies, Carlsbad, CA).

## Crystallization

16-Ac-KLVFFAENVGS-$NH_3$-26 (Aβ 16–26 D23N) was dissolved at 4.5 mg/ml in 20% DMSO. Micro crystals were grown in batch in 0.2M magnesium formate, 0.1M Tris base pH 8.0, and 15% isopropanol at room temperature under quiescent conditions. Crystals grew within 4 days to a maximum of 2 weeks.

## MicroED data collection

The procedures for MicroED data collection and processing largely follow published procedures (*Shi et al., 2016*; *Hattne et al., 2015*). Briefly, a 2–3 µl drop of crystals in suspension was deposited onto a Quantifoil holey-carbon EM grid then blotted and vitrified by plunging into liquid ethane using a Vitrobot Mark IV (FEI, Hillsboro, OR). Blotting times and forces were optimized to keep a desired concentration of crystals on the grid and to avoid damaging the crystals. Frozen grids were then either immediately transferred to liquid nitrogen for storage or placed into a Gatan 626 cryo-holder for imaging. Images and diffraction patterns were collected from crystals using FEI Tecnai 20 TEM with field emission gun (FEG) operating at 200 kV and fitted with a bottom mount TVIPS Tem-Cam-F416 CMOS-based camera. Diffraction patterns were recorded by operating the detector in a video mode using electronic rolling shutter with 2 × 2 pixel binning (*Nannenga et al., 2014*). Exposure times for these images were either 2 or 3 s per frame. During each exposure, crystals were continuously unidirectionally rotated within the electron beam at a fixed rate of 0.3 degrees per second, corresponding to a fixed angular wedge of 0.6 or 0.9 degrees per frame.

Crystals that appeared visually undistorted produced the best diffraction. Datasets from individual crystals were merged to improve completeness and redundancy. Each crystal dataset spanned a wedge of reciprocal space ranging from 40 to 80˚. We used a selected area aperture with an illuminating spot size of approximately 1 µm. The geometry detailed above equates to an electron dose rate of less than 0.01 e$^−$/Å$^2$ per second being deposited onto our crystals.

Measured diffraction images were converted from TIFF format into SMV crystallographic format, using publicly available software (available for download at http://cryoem.janelia.org/downloads).

We used XDS to index the diffraction images and XSCALE (*Kabsch, 2010*) for merging and scaling together datasets originating from thirteen different crystals.

## Structure determination

We determined the structure of Aβ 16–26 D23N using molecular replacement. KLVFFA (pdb 2Y2A) led us to our atomic model. The solution was obtained using Phaser (*McCoy, 2007*). Subsequent rounds of model building and refinement were carried out using COOT and Phenix, respectively (*Emsley and Cowtan, 2004*; *McCoy et al., 2005*). Electron scattering factors were used for refinement. Some reflections extended to 1.40 Å resolution. Calculations of the area buried and Sc were performed with AREAIMOL (*Collaborative Computational Project, 1994*; *Lee and Richards, 1971*) and Sc (*Connolly, 1983*; *Lawrence and Colman, 1993*; *Richards, 1977*), respectively.

## Computational structure-based design

Computational designs were carried out using the RosettaDesign software as described previously (*Sievers et al., 2011*). The atomic structure of the 16-KLVFFAENVGS-26 Aβ segment was used as a starting template for computational design. An extended L-peptide (or D-peptide, six to eight residues) was first placed at the end of the starting template of atomic structure. The design procedure then built side-chain rotamers of all residues onto the nine-residue peptide backbone placed at growing end of fibril. The optimal set of rotamers was identified as those that minimize an energy

function containing a Lennard-Jones potential, an orientation-dependent hydrogen bond potential, a solvation term, amino acid-dependent reference energies, and a statistical torsional potential that depends on the backbone and side-chain dihedral angles. Area buried and shape complementarity calculations were performed with areaimol and Sc, respectively, from the CCP4 suite of crystallographic programs (*Collaborative Computational Project, 1994*). The solubility of each peptide was evaluated by hydropathy index (*Kyte and Doolittle, 1982*). The designed peptides were selected based on calculated binding energy of top or bottom binding mode, shape complementarity and peptide solubility. Each structural model of selected peptides went through human inspection using Pymol, where those peptides with sequence redundancy and fewer binding interactions were omitted. Finally, select peptides were synthesized and tested experimentally.

## Sample preparation for electron microscopy

Aβ1–42 was dissolved and diluted as previously described. Inhibitor stocks were prepared in 100% DMSO and were added such that the sample contained 10 µM monomeric Aβ1–42 the indicated ratio of inhibitor with final concentration of 1% DMSO. Samples were incubated for 72 hr at 37°C under quiescent conditions. Aβ1–42 fibrils were formed as described, and then treated with indicated ration of inhibitor for 24 hr at 37°C under quiescent conditions. Fibril abundance was checked using electron microscopy.

## Transmission electron microscopy

Samples were spotted onto non-holey grids and left for 160 to 180 s. Remaining liquid was wicked off and then left to dry before analyzing. Samples for negative-stain TEM were treated with 2% uranyl acetate after sample was wicked off the grid. After 1 min, the uranyl acetate was wicked off. The grids were analyzed using a T12 Electron Microscope (FEI, Hillsboro, OR). Images were collected at 3200x or 24,000x magnification and recorded using a Gatan 2k × 2 k CCD camera.

## Thioflavin-T (ThT) kinetic assays

Thioflavin-T (ThT) assays were performed in black polystyrene 96-well plates (Nunc, Rochester, NY) or black polypropylene 96 well plates (Greiner Bio-One, Austria), as indicated, and sealed with UV optical tape. The total reaction volume was 150 µL per well. Aβ1–42 was prepared as described. Inhibitors were added at indicted concentrations, with a final concentration of 1% DMSO. ThT fluorescence was recorded with excitation and emission of 444 nm and 482 nm, respectively, using a Varioskan Flash (Thermo Fisher Scientific, Grand Island, NY). Experiments were performed at 37°C without shaking in triplicate and readings were recorded every 5 min. Seeding assay included 10% monomer equivalent of preformed fibrils, aggregated in LoBind polypropylene tubes, and sonicated for 10 min prior to addition. Inhibitor interruption assays were prepared as above in polypropylene plates. At approximal $T_{1/2}$, readings were paused and inhibitors were added as indicated, and plates were resealed with new UV optical tape.

ThT assays with tau40 were prepared as above with the following exceptions. 0.5 mg/mL heparin (Sigma cat. no. H3393) was added to the reaction mixture and experiments were performed at 37°C with double orbital shaking at 700 rpm. ThT assays with K18+ were prepared as above with the following exceptions. Experiments were performed at 37°C with double orbital shaking at 700 rpm, in polypropylene plates. Seeding assays included 10% monomer equivalent of preformed fibrils, sonicated for 10 min prior to addition.

## Cell culture

Neuro2a (N2a) cells (ATCC cat# CCL-131) were cultured in MEM media (Cat. # 11095–080, Life Technologies) plus 10% fetal bovine serum and 1% pen-strep (Life Technologies). Cells were cultured at 37°C in 5% CO2 incubator. Cells were authenticated by COX I gene analysis (Laragen), and mycoplasma negative by MycoAlert PLUS Detection Kit (Lonza, cat# LT07-701).

## 3-(4,5-dimethylthiazol-2-yl)−2,5-diphenyltetrazolium bromide (MTT) dye reduction assay for cell viability

N2a cells were plated at 5,000 cells per well in 90 µL of culture media, in clear 96-well plates (Cat. # 3596, Costar, Tewksbury, MA). Cells were allowed to adhere to the plate for 20–24 hr. Aβ1–42

samples were incubated at 10 µM with or without inhibitors at varying ratios for 12 hr at 37°C and then applied to N2a cells. 10 µL of sample was added to cells. By doing this, samples were diluted 1/10 from in vitro stocks. Experiments were done in triplicate.

After a 24-hr incubation, 20 µL of Thiazolyl Blue Tetrazolium Bromide MTT dye (Sigma, St. Louis, MO) was added to each well and incubated for 3.5 hr at 37°C under sterile conditions. The MTT dye stock is 5 mg/mL in Dulbecco's PBS. Next, the plate was removed from the incubator and the MTT assay was stopped by carefully aspirating off the culture media and adding 100 µL of 100% DMSO to each well. Absorbance was measured at 570 nm using a SpectraMax M5. A background reading was recorded at 700 nm and subsequently subtracted from the 570 nm value. Cells treated with vehicle alone (PBS+0.1% DMSO) were designated at 100% viable and cells treated with 100% DMSO designated as 0% viable, and cell viability of all other treatments was calculated accordingly. We employed one-way ANOVA as our statistical test for significance. Extended ANOVA data included as a supplementary file. $IC_{50}$ values were estimated using a four-parameter non-linear fit dose-response curve in Graphpad Prism.

## Dot Blot Assay

Aβ1–42 samples were incubated at 10 µM with or without inhibitors for 6, 24, and 72 hr at 37°C, and spotted onto a nitrocellulose membrane (Cat. # 162–0146, BioRad, Hercules, CA). 20 µL was loaded for each condition; 2 µL was spotted at a time and allowed to dry between application. The membranes were blotted as previously described (*Krotee et al., 2017*), with the exception of the primary antibodies used. The antibodies used in the assay were previously generated and characterized (*Hatami et al., 2014*). Blots were quantified with ImageJ.

## Surface Plasmon Resonance (SPR)

SPR experiments were performed using BiacoreT200 instrument (GE Healthcare). Aβ42 fibrils/tau K18 fibrils were immobilized on a CM5 sensor chip. The fibrils of Aβ42 were prepared by placing a sample of 50 µM Aβ42 in PBS pH 7.4 in two wells of a Nunc 96-well optical bottom plate (Thermo Scientific), 150 µl/well and incubating the plate in a microplate reader (FLUOstar Omega, BMG Lab-tech) at 37°C with double orbital shaking at 600 rpm overnight. Sample from the two wells were pooled together and Aβ42 fibrils were isolated from the incubation mixture by centrifuging it at 13,000 xG, 4°C for 45 min. The supernatant was removed and the pellet was re-dissolved in an equal volume of PBS as that of supernatant. The isolated fibrils were sonicated using a probe sonicator for 1–2 min at 18% amplitude with 2 s on, 5 s off pulses. The sonicated fibrils were filtered through a 0.22 µ filter to remove large aggregates. The sonicated and filtered fibrils were diluted to 60 µg/ml in 10 mM NaAc, pH 3 and then, immobilized immediately on a CM5 sensor chip using standard amine coupling chemistry. Briefly, the carboxyl groups on the sensor surface were activated by injecting 100 µl of 0.2 M EDC and 0.05 M NHS mixture over flow cells 1–2. The fibrils were then injected at a flow rate of 5 µl/min over flow cell 2 of the activated sensor surface for 900 s. The remaining activated groups in both the flow cells were blocked by injecting 120 µl of 1 M ethanol-amine-HCl pH 8. 5.. For the binding assay each peptide inhibitor was dissolved in 100% DMSO at a concentration of 1 mM and diluted in PBS pH 7.4+1.2% DMSO to concentrations ranging from 5 µM to 260 µM. Each peptide was injected at a flow rate of 30 µl/min over both flow cells (1 and 2) at increasing concentrations (in running buffer, PBS, pH 7.4+1.2% DMSO) at 25°C. For each sample the contact time and dissociation time were 120 s and 160 s, respectively. 3 M NaCl was used as regeneration buffer. The data were processed and analyzed using Biacore T200 evaluation software 3.1. The data of flow cell 1 (blank control) was subtracted from the data of flow cell 2 (with immobilized fibrils/monomers). The equilibrium dissociation constant (Kd) was calculated by fitting the plot of steady-state peptide binding levels (Req) against peptide concentration (C) with 1:1 binding model (*Equation 1*).

$$Req = \frac{CR_{max}}{Kd + C} + RI \tag{1}$$

$R_{max}$ = Analyte binding capacity of the surface
RI = Bulk refractive index contribution in the sample

## Recombinant Tau purification

K18, K18+, Human Tau40 (residues 1–441) WT, 3R and mutants: interface A (Q276W, L282R, I308P), interface B (Q276W, I277M, I308P), interface C (I277M, L282R, I308P)interface 1 (Q276W, I277M, L282R, Q307W, V309W), interface 2 (Q276W, I277M, L282R, I308W), were expressed in pET28b with a C-terminal His-tag (tau40) or or PNG2 (K18, K18+) in BL21-Gold *E. coli* cells grown in TB to an OD600 = 0.8. Cells were induced with 0.5 mM IPTG for 3 hr at 37°C and lysed by sonication in 50 mM Tris (pH 8.0) with 500 mM NaCl, 20 mM imidazole, 1 mM beta-mercaptoethanol, and HALT protease inhibitor. Cells were lysed by sonication, clarified by centrifugation at 15,000 rpm for 15 min, and passed over a 5 ml HisTrap affinity column. The column was washed with lysis buffer and eluted over a gradient of imidazole from 20 to 300 mM. Fractions containing purified Tau40 were dialyzed into 50 mM MES buffer (pH 6.0) with 50 mM NaCl and 1 mM beta-mercaptoethanol and purified by cation exchange. Peak fractions were polished on a HiLoad 16/600 Superdex 200 pg in 1X PBS (pH 7.4), and concentrated to ~20–60 mg/ml by ultrafiltration using a 10 kDa cutoff.

## Fibril incubation with inhibitors for tau biosensor cell-seeding assays

Aβ fibrils were prepared at 200 µM at 37°C for 72 hr before diluting to 50 µM in PBS buffer (pH 7.4) for seeding experiments. Tau40 WT and interface mutation fibrils were prepared by shaking 50 µM tau40 in PBS buffer (pH 7.4) with 0.5 mg/ml heparin (Sigma cat. no. H3393) and 1 mM dithiothreitol (DTT) for 3–6 days. Fibrillization was confirmed with an endpoint ThT reading, and fibrils were then diluted 20-fold to 1.25 µM in OptiMEM (Life Technologies, cat. no. 31985070). Inhibitors dissolved in DMSO were added to 20 µl of diluted fibrils at a concentration 20-fold greater than the final desired concentration. Fibrils were incubated for ~16 hr with the inhibitor, and subsequently were sonicated in a Cup Horn water bath for 3 min before seeding the cells. The resulting 'pre-capped fibrils' were mixed with one volume of Lipofectamine 2000 (Life Technologies, cat. no. 11668027) prepared by diluting 1 µl of Lipofectamine in 19 µl of OptiMEM. After 20 min, 10 µl of fibrils were added to 90 µl of the tau-K18CY biosensor cells to achieve the final indicated ligand concentration. Cells were verified by STR profiling and confirmed mycoplasma negative (Laragen). Quantification of seeding was determined by imaging the entire well of a 96-well plate seeded in triplicate and imaged using a Celigo Image Cytometer (Nexcelom) in the YFP channel. Aggregates were counted using ImageJ (*Eliceiri et al., 2012*) by subtracting the background fluorescence from unseeded cells and then counting the number of peaks with fluorescence above background using the built-in Particle Analyzer. We employed one-way ANOVA as our statistical test for significance. Extended ANOVA data included as a supplementary file. Dose-response curves were constructed for inhibitor peptides exhibiting concentration dependence by fitting to a nonlinear regression model in Graphpad Prism. High resolution images were acquired using a ZEISS Axio Observer D1 fluorescence microscope.

## Preparation of Brain lysate

Human brain tissue was obtained from the Neuropathology Laboratory at UCLA Medical Center. AD and PSP cases were confirmed by the Neuropathology Laboratory by immunostaining autopsied brain tissue sections, and the PSP donor was confirmed to be free of amyloid immunoreactivity. Tissue sections from the indicated brain regions were manually homogenized using a disposable ultra-tissue grinder (Thermo Fisher) in TBS (pH 7.4) supplemented with 1X HALT protease inhibitor. Homogenized tissue was aliquoted to several PCR tubes and prepared for seeding in biosensor cells by sonication as described by *Kaufman et al. (2017)*, except tissue sections were sonicated twice as long, for a total of 2 hr, in an ice cooled circulating water bath with individual sample tubes stirring to ensure each tube received the same sonication energy. Subsequently, seeding was measured by transfection into biosensor cells and quantified as described above. We employed one-way ANOVA as our statistical test for significance. Extended ANOVA data included as a supplementary file.

## Immunodepletion of Brain lysate

Lysate was prepared as above. 2 µg (0.2 µL at 11 µg/uL) Tau antibody (Dako A0024) was conjugated to 0.75 mg ProteinG Dynabeads (25 µL of 30 mg/mL). Antibody was mixed with beads and nutated for 10 min, washed with 200 µL Citrate-phosphate wash buffer pH 5.0, and then resuspended in a minimal volume of wash buffer. 200 µL of brain lysate diluted 1/20 in OptiMEM was added to

antibody-bead suspension and Nutated for 30 min. Supernatant was removed and used for transfection into biosensor cells, as previously described.

## Aggregation Inhibition Assay with α-synuclein

α-synuclein was expressed and purified as described previously in Rodriguez, et al. with the following exceptions to the expression protocol. An overnight starter culture was grown in 15 mL instead of 100 mL, 7 mL of which was used to inoculate 1 L. After induction, cells were allowed to grow for 3–4 hr at 34°C (instead of 4–6 hr at 30°C). Cells were then harvested by centrifuging at 5000 x g.

ThT assays with α-synuclein were performed in black 96-well plates (Nunc, Rochester, NY) sealed with UV optical tape. The total reaction volume was 180 μL per well. ThT fluorescence was recorded with excitation and emission of 444 nm and 482 nm, respectively, using a Varioskan Flash (Thermo Fisher Scientific, Grand Island, NY). Experiments were performed at 37°C, shaking at 600 rpm with a teflon bead, in triplicate and readings were recorded every 15 min. Alpha synuclein at 105 μM in PBS was diluted to a final concentration of 50 μM in 25 μM Thioflavin-T and PBS. Inhibitors were added at the specified concentration by diluting 10 mM stocks in 100% DMSO 1 to 40 in the same manner. Thus, inhibitors were tested at 5:1 molar excess of α-synuclein.

## Aggregation Inhibition Assay with IAPP

Human IAPP1-37NH$_2$ (hIAPP) was purchased for Innopep (San Diego, CA). Peptides were prepared by dissolving lyophilized peptide in 100% 1,1,1,3,3,3-Hexafluoro-2-propanol (HFIP) at 250 μM for 2 hr. Next, the sample was spin-filtered and then HFIP was removed with a CentriVap Concentrator (Labconco, Kansas City, MO). After removal of the HFIP, the peptide was dissolved at 1 mM or 10 mM in 100% DMSO (IAPP alone) or 100% DMSO solutions containing 1 mM or 10 mM inhibitor. The DMSO peptide stocks were diluted 100-fold in filter-sterilized Dulbecco's PBS (Cat. # 14200–075, Life Technologies, Carlsbad, CA). Thioflavin-T (ThT) assays with hIAPP were performed in black 96-well plates (Nunc, Rochester, NY) sealed with UV optical tape. hIAPP1-37NH$_2$ and mIAPP1-37NH$_2$ were prepared as described. The total reaction volume was 150 μL per well. ThT fluorescence was recorded with excitation and emission of 444 nm and 482 nm, respectively, using a Varioskan Flash (Thermo Fisher Scientific, Grand Island, NY). Experiments were performed at 25°C without shaking in triplicate and readings were recorded every 5 min.

## Atomic structure overlay

A structural superposition of Aβ 16–26 and tau (5V5B, 6HRE) was performed using LSQ from coot (*Emsley and Cowtan, 2004*). We calculated root mean square deviation (RMSD) of main chains for parallel orientations fitting 6–8 residues. Anti-parallel LSQ computation of Aβ 16–22 and tau 275–281 (5V5B) of C$_\alpha$ atoms was calculated, and side chain rotamers optimized with Foldit (*Kleffner et al., 2017*) over 2000 iterations to minimize energy to −603 REU. For side seeding model, residues 16–21 of Aβ (5OQV) were superimposed on 304–309 of Tau (6HRF). Tau was then manually moved perpendicular to the fibril axis to make a complementary surface with 5OQV. Backbone and side chain rotamers were optimized with Foldit to minimize energy to −1517REU.

## Acknowledgements

We thank M Diamond for gifting the monoclonal biosensor HEK293 cell-line that expressed tau-K18 (P301S) EYFP for our inhibitor assay. We thank Dr. Vinters and Christopher K Williams for supplying patient tissue and immunohistological summaries of tissues. We thank Dan Anderson for general support in the laboratory. We thank Lorena Saelices for providing TTR fibers and Qin Cao for providing TDP-43 fibers. We thank Lin Jiang for helpful conversation regarding Aβ toxicity. We thank the UCLA-DOE X-ray Crystallography Core Technology Center; the Janelia Research Campus visitor program and Ivo Atanasov and the Electron Imaging Center for NanoMachines (EICN) of California NanoSystems Institute (CNSI) at UCLA for the use of their electron microscopes. We thank Johan Hattne for assistance processing MicroED data. The UCLA-DOE X-ray Crystallization Core Technology Center is supported in part by the Department of Energy grant DE-FC0302ER63421. The Gonen laboratory is funded by the Howard Hughes Medical Institute.

# Additional information

## Competing interests

David S Eisenberg: is a SAB member and equity holder in ADRx, Inc. The other authors declare that no competing interests exist.

## Funding

| Funder | Grant reference number | Author |
| --- | --- | --- |
| National Institutes of Health | R01 AG029430 | Sarah L Griner<br>Paul Seidler<br>Jeannette Bowler<br>Kevin A Murray<br>Tianxiao Peter Yang<br>Shruti Sahay<br>Michael R Sawaya<br>Duilio Cascio<br>Jose A Rodriguez<br>David S Eisenberg |
| Howard Hughes Medical Institute | | Sarah L Griner<br>Paul Seidler<br>Jeannette Bowler<br>Kevin A Murray<br>Tianxiao Peter Yang<br>Shruti Sahay<br>Michael R Sawaya<br>Duilio Cascio<br>Jose A Rodriguez<br>Tamir Gonen<br>David S Eisenberg |
| Cure Alzheimer's Fund | | Stephan Philipp<br>Justyna Sosna<br>Charles G Glabe |
| National Institutes of Health | R56 AG061847 | Paul Seidler |
| National Institutes of Health | R01 AG054022 | Sarah L Griner<br>Paul Seidler<br>Jeannette Bowler<br>Kevin A Murray<br>Tianxiao Peter Yang<br>Shruti Sahay<br>Michael R Sawaya<br>Duilio Cascio<br>David S Eisenberg |

The funders had no role in study design, data collection and interpretation, or the decision to submit the work for publication.

## Author contributions

Sarah L Griner, Conceptualization, Formal analysis, Validation, Investigation, Visualization, Writing—original draft, Project administration; Paul Seidler, Formal analysis, Validation, Investigation, Visualization, Writing—review and editing; Jeannette Bowler, Formal analysis, Validation, Investigation, Writing—review and editing; Kevin A Murray, Software, Formal analysis, Investigation; Tianxiao Peter Yang, Formal analysis, Investigation; Shruti Sahay, Jose A Rodriguez, Formal analysis, Investigation, Visualization; Michael R Sawaya, Software, Formal analysis, Writing—review and editing; Duilio Cascio, Software, Formal analysis; Stephan Philipp, Resources, Investigation, Visualization; Justyna Sosna, Resources, Investigation; Charles G Glabe, Resources, Supervision, Methodology; Tamir Gonen, Resources, Formal analysis, Investigation; David S Eisenberg, Conceptualization, Resources, Supervision, Funding acquisition, Writing—review and editing

## Author ORCIDs
Sarah L Griner (ID) https://orcid.org/0000-0002-4500-2419
Jeannette Bowler (ID) https://orcid.org/0000-0003-2428-3095
Tianxiao Peter Yang (ID) http://orcid.org/0000-0002-4479-5154
Tamir Gonen (ID) http://orcid.org/0000-0002-9254-4069
David S Eisenberg (ID) https://orcid.org/0000-0003-2432-5419

## Decision letter and Author response
Decision letter https://doi.org/10.7554/eLife.46924.035
Author response https://doi.org/10.7554/eLife.46924.036

## Additional files

### Supplementary files
• Source data 1. Extended ANOVA data.
DOI: https://doi.org/10.7554/eLife.46924.029

• Supplementary file 1. Computed binding properties of designed inhibitors to Aβ $_{16}$KLVFFAEN$_{23}$.
DOI: https://doi.org/10.7554/eLife.46924.030

• Transparent reporting form DOI: https://doi.org/10.7554/eLife.46924.031

### Data availability
Diffraction data have been deposited in PDB under the accession code 6O4J. Source data for toxicity and seeding data are provided (Figures 2–7).

The following dataset was generated:

| Author(s) | Year | Dataset title | Dataset URL | Database and Identifier |
|---|---|---|---|---|
| Griner SL, Sawaya MR, Rodriguez JA, Cascio D, Gonen T | 2019 | Amyloid Beta KLVFFAENVGS 16-26 D23N Iowa mutation | https://www.rcsb.org/structure/6O4J | Protein Data Bank, 6O4J |

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
