## [Decision Letter]

Thank you for submitting your article "Structure based inhibitors of Amyloid Beta core suggest a common interface with Tau" for consideration by *eLife*. Your article has been reviewed by three peer reviewers, and the evaluation has been overseen by a Reviewing Editor and Cynthia Wolberger as the Senior Editor. The following individuals involved in review of your submission have agreed to reveal their identity: Sara Linse (Reviewer #1).

The reviewers have discussed the reviews with one another and the Reviewing Editor has drafted this decision to help you prepare a revised submission.

The authors report the structure of D23N-Ab(16-26) in an antiparallel cross-β form, design and optimize peptide-based inhibitors based on this structure, and demonstrate inhibition of Aβ aggregation, cross-seeding of tau aggregation by Aβ, and tau aggregation itself. The work is exciting because it suggests that Aβ and tau share a common structural feature responsible for both Aβ toxicity and cross-seeding of tau, which could directly link early Aβ aggregation to late tau aggregation – a major question in how aggregation is linked to degeneration in Alzheimer's.

The work is fundamentally important and a revised manuscript can be appropriate for *eLife*, but there are issues that need to be addressed in order for the work to reach to the *eLife* standard of reliability and impact. As summarized below, these fall into three categories: 1) Establishing general reliability of the assembly reactions performed in the presence of a polystyrene surface. 2) Explaining better the logic of using an antiparallel cross-β structure as the basis for designing inhibitors of Aβ2 fibrils, which have parallel structures. 3) Concerns about technical aspects of the study and the extent to which different aspects of the system have been characterized adequately.

Significant issues that need to be addressed:

1) Influence of the reaction matrix.

Please explain why Aβ_1-42_ aggregation is so slow and specify the surface material of the 96 well. If this is plain polystyrene then the reaction could be dominated by the effect of polystyrene and that it is this effect that the inhibitors are interfering with (polystyrene is avoided in most current studies as it either catalyzes or retards aggregation depending on peptide concentration to surface area ratio and even makes the reaction concentration-independent with the polystyrene surface being limiting for the reaction rate). The basic kinetic measurements should be repeated in non-binding plates. Related issues include:

i) The small effect of 10% Aβ_1-42_ seed (Figure 5—figure supplement 1). Is this an artefact of polystyrene? ii) Please specify the container in which Aβ or Aβ plus inhibitor was incubated and the surface material used during aggregation reactions for other proteins. Could we be looking here at polystyrene-catalyzed aggregation?

2) Inhibitor design strategy

Please explain earlier in the text the rational for using an antiparallel cross-β structure as the basis for developing inhibitors. The inhibitors appear to inhibit the formation of Aβ42 fibrils, which have parallel cross-β structures. Why, if the inhibitors were designed to cap an antiparallel structure? [In related work, Wei Qiang recently showed that structure-based inhibitors could direct D23N-Ab(1-40) self-assembly toward either antiparallel or parallel structures (see J. Phys. Chem. B, vol. 121, p. 5544, 2017)]

3) System characterization

i) Aggregation of tau is studied both in vitro and in a cellular model system. In none of these conditions was the phosphorylation status of tau determined/discussed. Certainly, in the in vitro conditions employed there is no phosphorylation of tau. Given that tau phosphorylation is a hallmark of tau neurofibrillary tangles, it is highly likely that both the biophysical properties and interactions of tau will be highly coupled with its phosphorylation status. Therefore, it is imperative to evaluate whether and how the phosphorylation status of tau affects inhibitor activity.

ii) The authors try to establish whether inhibitors affect oligomer formation by the use of antibodies against specific oligomeric/fibrillar conformations. An effect of the inhibitors can only be appreciated at a 1:10 molar ratio of Aβ:inhibitor. Yet, the ThT data suggest that inhibitors are very effective at 1:1 molar ratio (reduction of more than 80% of ThT signal amplitude). Only one of the antibodies used (A11-O9) probes oligomers. All the other antibodies probe against fibrillar assemblies. These last antibodies are expected not to work (ThT is negative in the presence of inhibitors). Therefore, the whole argument that oligomer formation is being affected is based on a single antibody western-blot result. There is no quantification of these results and the authors should specify how many times the experiment was performed. There are commercially available dyes that detect oligomeric assemblies of Aβ, which the authors should use to validate their single observation.

iii) The authors attempt to establish whether inhibitors cap fibril ends as predicted by their design or if they disassemble fibrils. The experiment is to add inhibitors at 72 hours post fibrillation begins. The analysis is performed through negative stain TEM. The result shows no difference between control or inhibitor treated samples (Figure 4B). It is then assumed that inhibitors cap or coat fibrils. A better experiment would be to add the inhibitors during exponential phase of fibril formation. In this case, if the hypothesis is true, fibril elongation should stop and this could be analyzed using fibril yield/ThT/AFM or negative stain TEM analysis of fibril length distribution.

iv) Subsection “Inhibitors bind and reduce toxicity of Aβ aggregates”: the authors report the Kd value for D1d inhibitor binding to Aβ fibrils. The IC50 value for toxicity reduction of Aβ fibrils with the same inhibitor is about 50 times lower. This makes it difficult to rationalize how 50% of activity is acquired at such small fraction of inhibitor bound given the reported Kd.

v) Subsection “Inhibitors reduce tau aggregation and seeding”: the effect of mutagenesis is not evaluated over propensity of mutated tau to form fibrils. In fact, not all mutants seed with the same efficiency. The authors should provide evidence (negative stain TEM, fibril yield) that tau fibrils are still formed so as to demonstrate that the effects observed are due to specific alterations of binding epitopes and not to a general effect over fibril formation caused by the mutations.

vi) Several elements in this experiment need to be controlled. First, the amount of tau aggregate load might not be equal in the different tissues. This would explain the different seeding efficiencies observed (Figure 7A) and would allow the results to be interpreted in terms of disease progression correlated with tau deposit load. An immuno-depletion control of tau is necessary to support a claim for a direct effect of inhibitors over tau and tau-mediated seeding.

Issues for the authors' consideration

1) Please describe the starting state for the ThT assays? How was monomer isolated?

2) How does the D23N-Aβ(16-26) crystal structure compare with the antiparallel D23N-Aβ(1-40) fibril structure described in reference 32? A brief discussion may be appropriate. The hydrogen-bond registry appears to be different.

3) In Figure 4A, are the pre-formed Aβ42 assemblies fibrils, or are they non-fibrillar oligomers? The text is ambiguous about this point.

4) The Discussion makes that point that structures of tau and Aβ fibrils are similar, based on crystal structures of tau peptides and Aβ peptides (Figure 6). These crystal structures are indeed similar, but are the structures of fibrils formed by full-length Aβ and tau also similar? Multiple fibril structures are now available from solid state NMR and cryoEM, so it would make sense to compare these fibril structures directly because fragments beyond the amyloid core apparently play an important role in modulating the final fibril conformation achieved. This makes it very difficult (if not dangerous) to make conclusions about the behavior of effects of Aβ of physiological length with the conformation observed in this work for Aβ16-26-D23N. At least this should be pointed out as a potential major caveat of the work.

5) Is the effect of the inhibitors on seeding of tau aggregation by Aβ fibrils due to capping or coating of the Aβ fibrils, or is it due to direct inhibition of tau aggregation (which the authors also demonstrate)?

6) The paper concludes with the idea that "a direct interaction between the Aβ core and the amyloid-prone regions of tau facilitates cross-seeding". Can the authors be more specific about what this interaction is and how it facilitates cross-seeding, and about how their data support this idea? Do tau fibrils grow from the ends of Aβ fibrils? Or on the sides of Aβ fibrils? Or is a non-fibrillar state of Aβ important? Or perhaps a non-parallel structure that exists only at the ends of Aβ fibrils? Or perhaps tau oligomers form on the ends or sides of Aβ fibrils, and then tau fibrils nucleate in these tau oligomers?

7) The work does not provide direct evidence that a common surface or motif exists in Aβ, as was found for tau. This could have been done simply by mutating Aβ, in the way that tau was mutated in this study and repeating the Ab seeding study of tau.

8) The conclusions of this study are based on the development of peptide inhibitors that the authors demonstrate affect the formation of Aβ fibrils, with concomitant effects on toxicity and seeding. However, there is no assessment of the concentration of monomer or oligomers that remain in solution: these would be expected to increase as a consequence of impaired fibrillation. Such data would help to understand the mode of action of the presented inhibitors.

9) Why is the crystalline arrangement of Aβ16-26 peptide described in this work considered to be a fibril? It is no different from previous steric zippers that are understood to be in a crystalline arrangement, such as those already described by the same group as steric zippers Class 7.

10) Subsection “Efficacy of inhibitors of Aβ aggregation designed against Aβ 16-26 D23N”: please specify how IC50 was estimated.

11) Subsection “Reduction of toxicity by designed inhibitors is explained by a reduction of Aβ_1-42_ aggregation”: The use of negative stain TEM to demonstrate the absence of fibrils is prone to over-interpretation. A more quantitative method, such as fibril yield (or soluble species yield) by centrifugation would strengthen the authors' argument.

12) A related issue is that there is a general lack of characterization of which particles are induced by interaction with inhibitor. DLS data would help to make the point regarding which species (monomers, oligomers, small fibrils) the inhibitor is binding to.

13) A better use of Kd value for D1d inhibitor binding to Aβ fibrils would be to compare it with the Kd of D1b peptide which is less active than D1d. Also, there are other experimental methodologies such as microscale thermophoresis that have been reported to cope better with self-aggregation of the analytes that the authors claim to disturb their SPR based binding experiments.

14) Subsection “Inhibitors reduce seeding of tau by aggregated Aβ_1-42_”: the authors claim that in their experiments, fibrils of Aβ_1-42_ seed tau aggregation as efficiently as fibrils from tau K18. Yet the results don't show a clear reduction on Tlag (Figure 5A) that is expected and clearly observed in the Aβ_1-42_ self-seeding (Figure 5—figure supplement 1, panel B)

15) Subsection “Inhibitors reduce seeding of tau by aggregated Aβ_1-42_”: the authors imply that fibrils from hIAPP, TDP43, α-synuclein, and TTR are morphologically similar. This is incorrect.

16) Subsection “Inhibitors reduce seeding of tau by aggregated Aβ_1-42_”: the observed inhibition of Aβ_1-42_ fibril seeding of tau aggregation seems not to be dose dependent, except for inhibitor D1b. More experimental points in the dose response to inhibitors should be provided (Figure 5D).

17) Subsection “Inhibitors reduce tau aggregation and seeding”: control LC peptide (L-amino acid version of D1 peptide) does increase aggregation in vitro (reduced Tlag in ThT) and in cellular assays (increased number of seeded aggregates in tau-K18 biosensor cells). Can the authors comment how the stereochemistry of the peptides influences aggregation induction/inhibition?

18) Discussion paragraph four: in this work there is no characterization of the aggregation pathways of Aβ and tau. It thus is inappropriate to suggest that both molecules share a common aggregation pathway based on responsiveness to a peptide inhibitor.

---

## [Author Response]

The authors report the structure of D23N-Ab(16-26) in an antiparallel cross-β form, design and optimize peptide-based inhibitors based on this structure, and demonstrate inhibition of Aβ aggregation, cross-seeding of tau aggregation by Aβ, and tau aggregation itself. The work is exciting because it suggests that Aβ and tau share a common structural feature responsible for both Aβ toxicity and cross-seeding of tau, which could directly link early Aβ aggregation to late tau aggregation – a major question in how aggregation is linked to degeneration in Alzheimer's.The work is fundamentally important and a revised manuscript can be appropriate for eLife, but there are issues that need to be addressed in order for the work to reach to the eLife standard of reliability and impact. As summarized below, these fall into three categories: 1) Establishing general reliability of the assembly reactions performed in the presence of a polystyrene surface. 2) Explaining better the logic of using an antiparallel cross-β structure as the basis for designing inhibitors of Aβ2 fibrils, which have parallel structures. 3) Concerns about technical aspects of the study and the extent to which different aspects of the system have been characterized adequately.Significant issues that need to be addressed:1) Influence of the reaction matrix.Please explain why Aβ_1-42_ aggregation is so slow and specify the surface material of the 96 well. If this is plain polystyrene then the reaction could be dominated by the effect of polystyrene and that it is this effect that the inhibitors are interfering with (polystyrene is avoided in most current studies as it either catalyzes or retards aggregation depending on peptide concentration to surface area ratio and even makes the reaction concentration-independent with the polystyrene surface being limiting for the reaction rate). The basic kinetic measurements should be repeated in non-binding plates. Related issues include:i) The small effect of 10% Aβ_1-42_ seed (Figure 5—figure supplement 1). Is this an artefact of polystyrene? ii) Please specify the container in which Aβ or Aβ plus inhibitor was incubated and the surface material used during aggregation reactions for other proteins. Could we be looking here at polystyrene-catalyzed aggregation?

We thank the reviewers for alerting us to the possible interactions of polystyrene with Aβ. Our original kinetic measurements were performed in polystyrene plates, and the Materials and methods section has been corrected to state this. As suggested, to ensure that the effect of our inhibitors on Aβ was not influenced by the plate material, we repeated the experiments as shown in Figure 3A and Figure 5—figure supplement 1. We found that the lag time was decreased to ~8 hours in the polypropylene plates from around 11.5 hours in the polystyrene plates. The inhibition results were similar using polypropylene plates to the results from experiments performed in polystyrene plates, as displayed in Figure 3—figure supplement 1A. As the inhibition was consistent regardless of plate type, we did not repeat the measurements for aggregation of our other proteins (tau, hIAPP, α-synuclein). All additional experiments requested by reviewers were performed in polypropylene plates.

We are not overly concerned regarding the long lag time in our assays. We have found that Aβ aggregation lag times are quite variable based on the purification method, as well methodology of the assay, as time point readings perturb the plate and increases aggregation rate. Variable lag times from minutes to days have been reported in the literature for Aβ42 at concentrations that are comparable to those used in our experiments. Our recombinant preparation of Aβ yields reproducible lag times across preparations (see Krotee et al., 2017). We do not remove salts, such as TFA, from HPLC purification, so the inclusion of these in our starting material could be influencing the lag times in comparison the faster aggregation rates of SEC isolated monomers, in which have these salts have been removed.

2) Inhibitor design strategyPlease explain earlier in the text the rational for using an antiparallel cross-β structure as the basis for developing inhibitors. The inhibitors appear to inhibit the formation of Aβ42 fibrils, which have parallel cross-β structures. Why, if the inhibitors were designed to cap an antiparallel structure? [In related work, Wei Qiang recently showed that structure-based inhibitors could direct D23N-Ab(1-40) self-assembly toward either antiparallel or parallel structures (see J. Phys. Chem. B, vol. 121, p. 5544, 2017)]

We have amended the text in the Results section to further clarify our rational for inhibitor design. We have edited the text in the Results section (subsection “Atomic structure of Aβ16-26 D23N determined using MicroED” paragraph two) to include the 1-42 structure determined by cryoEM (5OQV), as a way to better explain how the inhibition of Aβ42 fibrils could occur by targetingAβ 16-26. We have added a structural overlay of Aβ16-26 and Aβ42 fibrils in Figure 1—figure supplement 2. Whereas we used the antiparallel structure to design the inhibitors, our goal was to inhibit a common structural motif that occurs in variety of larger assemblies, both antiparallel and parallel.

3) System characterization

*i) Aggregation of tau is studied both* in vitro *and in a cellular model system. In none of these conditions was the phosphorylation status of tau determined/discussed. Certainly, in the* in vitro *conditions employed there is no phosphorylation of tau. Given that tau phosphorylation is a hallmark of tau neurofibrillary tangles, it is highly likely that both the biophysical properties and interactions of tau will be highly coupled with its phosphorylation status. Therefore, it is imperative to evaluate whether and how the phosphorylation status of tau affects inhibitor activity.*

It is true that we do not explicitly determine the phosphorylation status of tau, but we make every effort to consider it in the most relevant possible context, by testing the effect of D1b on seeding by human brain derived tau fibrils in biosensor cells, which are an established model of tau aggregation and provide an environment that is conducive to tau phosphorylation.

We argue that the explicit characterization of the different possible post-translational modifications of tau from diseased tissues is beyond the scope for our manuscript, because such an analysis would require tissues from numerous other tau donors to be rigorously scrutinized with a panel of antibodies that are specific for different post-translational modifications, and/or mass spectrometry. Furthermore, such analysis would still be speculative, since it is not even established if or how post-translational modifications alter the biophysical properties and interactions of tau fibrils, and whether post-translational modifications of tau are even known comprehensively, or the frequency of those that have been observed.

Most importantly, the majority of AD related phosphorylation sites identified thus far occur outside of the K18 region. In the K18 region, phosphorylation of amino acids 258,262, 289 and 356 is observed in AD derived tau, (Martin, L., Latypova, X., and Terro, F. (2011) Post-translational modifications of tau protein: Implications for Alzheimer’s disease. Neurochemistry International. 58, 458–471; Duka, V., Lee, J. H., Credle, J., Wills, J., Oaks, A., Smolinsky, C., Shah, K., Mash, D. C., Masliah, E., and Sidhu, A. (2013) Identification of the Sites of Tau Hyperphosphorylation and Activation of Tau Kinases in Synucleinopathies and Alzheimer’s Diseases. PLoS ONE. 8, 1–11) none of which lie in the sequence regions we have identified for inhibitor binding. Furthermore, phosphate groups or other PTMs are not seen in the near-atomic structures of tau fibrils, which were solved at resolutions where it is possible to recognize density for such modifications. (Kenner, L. R., Anand, A. A., Nguyen, H. C., Myasnikov, A. G., Klose, C. J., McGeever, L. A., Tsai, J. C., Miller-Vedam, L. E., Walter, P., and Frost, A. (2019) eIF2B-catalyzed nucleotide exchange and phosphoregulation by the integrated stress response. Science. 364, 491 LP – 495). We suggest that although hyperphosphorylation may play a role in the tau aggregation pathway (monomer to aggregate), it likely does not play a major structural role in stabilizing the fibril core, and that the binding of the D1b inhibitor does not directly rely on any given PTM.

Therefore, we find it most prudent to limit our analysis to immunopathologically confirmed cases of tau pathology, and to avoid speculating about how post-translational modifications might affect the properties of seeding and inhibition. We add to our discussion a statement pointing out the caveat that “different co-factors and/or post-translational modifications, such as tau phosphorylation, could affect the ability of Aβ to seed tau, and the efficacy of D1b on tau seeding”, as cited in conjunction to our response to an issue that was raised below in point #4.

ii) The authors try to establish whether inhibitors affect oligomer formation by the use of antibodies against specific oligomeric/fibrillar conformations. An effect of the inhibitors can only be appreciated at a 1:10 molar ratio of Aβ:inhibitor. Yet, the ThT data suggest that inhibitors are very effective at 1:1 molar ratio (reduction of more than 80% of ThT signal amplitude). Only one of the antibodies used (A11-O9) probes oligomers. All the other antibodies probe against fibrillar assemblies. These last antibodies are expected not to work (ThT is negative in the presence of inhibitors). Therefore, the whole argument that oligomer formation is being affected is based on a single antibody western-blot result. There is no quantification of these results and the authors should specify how many times the experiment was performed. There are commercially available dyes that detect oligomeric assemblies of Aβ, which the authors should use to validate their single observation.

We thank the reviewers for this observation, and agree that reduction of oligomer formation is not adequately characterized. We have repeated our dot blot experiment and quantified the results of A11 (polyclonal) binding to Aβ samples incubated with inhibitor, included as Figure 3—figure supplement 2. We have also added a line in the text (Subsection “Reduction of toxicity by designed inhibitors is explained by a reduction of Aβ_1-42_ aggregation” paragraph two) stating that we have not determined what distinct oligomeric assemblies are being reduced by addition of our inhibitors.

We tried to analyze oligomer formation using the dye bis-ANS, which detects hydrophobicity and is commonly used to monitor oligomer formation. Other oligomer specific dyes such a as BD-oligo and other modified BODIPY molecules are not commercially available yet. We found that bis-ANS produced a very strong signal with our inhibitors alone, possibly due to their hydrophobic character and self-association, as well as their limited solubility necessitating preparation in DMSO. Thus, this dye could not be used to monitor Aβ oligomer formation with addition of the inhibitors.

iii) The authors attempt to establish whether inhibitors cap fibril ends as predicted by their design or if they disassemble fibrils. The experiment is to add inhibitors at 72 hours post fibrillation begins. The analysis is performed through negative stain TEM. The result shows no difference between control or inhibitor treated samples (Figure 4B). It is then assumed that inhibitors cap or coat fibrils. A better experiment would be to add the inhibitors during exponential phase of fibril formation. In this case, if the hypothesis is true, fibril elongation should stop and this could be analyzed using fibril yield/ThT/AFM or negative stain TEM analysis of fibril length distribution.

We thank the reviewers for this suggestion. The primary purpose of the experiment in the Figure 4B was to show that the fibril structures are not dissolved, as this could be one mechanism for inhibition of toxicity and seeding. We have performed the suggested experiment and found that addition of inhibitors at 1:1 completely stops fibril growth, and at 10 Aβ:1inhibitor can slow fibril growth as monitored by ThT, and included this as Figure 4—figure supplement 1B. We are cautious not to overinterpret the result of this experiment, as the inhibitor could feasibly be sequestering free monomer or small assemblies from adding to the fibrils and not just capping the fibrils already present and have noted this in the text (subsection “Inhibitors bind and reduce toxicity of Aβ aggregates” paragraph two).

iv) Subsection “Inhibitors bind and reduce toxicity of Aβ aggregates”: the authors report the Kd value for D1d inhibitor binding to Aβ fibrils. The IC50 value for toxicity reduction of Aβ fibrils with the same inhibitor is about 50 times lower. This makes it difficult to rationalize how 50% of activity is acquired at such small fraction of inhibitor bound given the reported Kd.

We do report an IC50 that is around 1μM for when the inhibitors are added to monomer, which prevents fibrillization. The Kd we report is for the inhibitor D1d binding to fibrillar Aβ. A better experiment to compare to our reported Kd is our experiment of fibrillar Aβ seeding tau biosensor cells (Figure 5C), or Aβ aggregates treated with inhibitor on N2a cells. We report the IC50s for both of these processes to be at a minimum ~5μM, or ~10 times lower than our reported Kd. We also see a variability in D1d efficacy on pre-aggregated structures, with it being a better inhibitor for oligomeric toxicity rather than seeding from fibrils, the process for which our experimental Kd is more applicable. This is not an uncommon trend: peptide inhibitors are much better at preventing aggregation of the monomer than of reducing toxicity of already formed aggregates.

v) Subsection “Inhibitors reduce tau aggregation and seeding”: the effect of mutagenesis is not evaluated over propensity of mutated tau to form fibrils. In fact, not all mutants seed with the same efficiency. The authors should provide evidence (negative stain TEM, fibril yield) that tau fibrils are still formed so as to demonstrate that the effects observed are due to specific alterations of binding epitopes and not to a general effect over fibril formation caused by the mutations.

We agree with the reviewers that we have not presented sufficient evidence of fibrillization of our mutants. To provide evidence, we include in our revised manuscript the characterization of the mutated tau by ThT and EM in Figure 6—figure supplement 2. The mutants do cause differences in maximal ThT signal, which is not correlated to seeding efficiency. It could be due to altered binding of ThT to the different fibril structures being formed, rather than differences in fibril load. Nevertheless, these data confirm that tau fibrils are still formed for each mutant.

vi) Several elements in this experiment need to be controlled. First, the amount of tau aggregate load might not be equal in the different tissues. This would explain the different seeding efficiencies observed (Figure 7A) and would allow the results to be interpreted in terms of disease progression correlated with tau deposit load. An immuno-depletion control of tau is necessary to support a claim for a direct effect of inhibitors over tau and tau-mediated seeding.

We agree the tau aggregate load probably is not equal in the different tissues, and this is likely the reason for different seeding efficiencies that are observed from different tissue types; we have noted this in the revised text (subsection “Designed inhibitor D1b targets disease relevant conformations” paragraph two). However, our point is not that D1b works differently at different stages of disease progression, but simply that D1b reduces tau seeding by tissue from each of the various regions sampled, and could be of use treating patients of various stages of the disease. We merely point out that seeding power, in this case, corresponds to the pattern expected based on Braak staging. The hippocampus, which is affected prior to the neocortical regions including the frontal and occipital lobes, could thus have more aggregated tau. We do observe stronger seeding from the hippocampal section and this could explain the relative reduced fractional efficacy of D1b.

We have included seeding data for both a control non-diseased patient brain, as well as an immune-depleted PSP brain in Figure 7A to confirm that misfolded tau from the tissue mediates seeding.

Issues for the authors' consideration1) Please describe the starting state for the ThT assays? How was monomer isolated?

Aβ for ThT was prepared by resuspending lyophilized Aβ from HPLC purification in NH_4_OH and filtering as described in the Materials and methods section. We did not isolate the monomer by SEC prior to ThT assays.

2) How does the D23N-Aβ(16-26) crystal structure compare with the antiparallel D23N-Aβ(1-40) fibril structure described in reference 32? A brief discussion may be appropriate. The hydrogen-bond registry appears to be different.

Whereas both D23N-Aβ(16-26) and D23N-Aβ(1-40) fibril are antiparallel structures, the former stacks with β sheets out of register, the latter is in register, which we have now described in the text (subsection “Atomic structure of Aβ16-26 D23N determined using MicroED”. So yes, the hydrogen bond registry is different. It is this registration difference that makes the D23N-Aβ(16-26) structure have similarities to oligomers, rather than just fibrils.

3) In Figure 4A, are the pre-formed Aβ42 assemblies fibrils, or are they non-fibrillar oligomers? The text is ambiguous about this point.

We have added in the revised manuscript a dot blot (A11, OC, 6E10) to show the species used in the sample in Figure 4A as Figure 4—figure supplement 1. The preformed aggregates are primarily non-fibrillar oligomers, but some fibrillar aggregates are present.

4) The Discussion makes that point that structures of tau and Aβ fibrils are similar, based on crystal structures of tau peptides and Aβ peptides (Figure 6). These crystal structures are indeed similar, but are the structures of fibrils formed by full-length Aβ and tau also similar? Multiple fibril structures are now available from solid state NMR and cryoEM, so it would make sense to compare these fibril structures directly because fragments beyond the amyloid core apparently play an important role in modulating the final fibril conformation achieved. This makes it very difficult (if not dangerous) to make conclusions about the behavior of effects of Aβ of physiological length with the conformation observed in this work for Aβ16-26-D23N. At least this should be pointed out as a potential major caveat of the work.

We agree that the smaller peptide structures do not always reflect the structures of full-length fibrils. However, while the inhibitors were designed against the Aβ16-26-D23N segment, all further characterization on their efficacy was done using Aβ _1-42_. We have added to the revision a comparison of β16-26-D23N segment to full length structure determined by cryoEM, showing the similarity to the region used for inhibitor design in subsection “Atomic structure of Aβ16-26 D23N determined using MicroED”.

We have amended the Discussion (paragraph four) to make it clear that the overall structure of full-length fibrils of Aβ and tau are not similar, but rather that they contain surfaces that are similar. We have added a model of Aβ _1-42_ and tau [Figure 6—figure supplement 2], in which we highlight an interface that could facilitate cross-seeding and be targeted by our inhibitor.

We also have pointed out (Discussion paragraph five) that it is possible that D1b could be efficacious for only some subtypes of aggregated tau, and that there may exist other fibril polymorphs with different structural conformations that might not be sensitive to D1b inhibition. Moreover, it is not known how co-factors and post-translational modifications, such as phosphorylation, might affect seeding of tau by Aβ, or D1b inhibitor efficacy. Thus, Aβ cross-seeding may represent one of many possible different structural stimuli of tau aggregation.

5) Is the effect of the inhibitors on seeding of tau aggregation by Aβ fibrils due to capping or coating of the Aβ fibrils, or is it due to direct inhibition of tau aggregation (which the authors also demonstrate)?

We cannot rule out that excess inhibitor could be interacting with tau monomer, as we do see inhibition of tau monomer aggregation induced with heparin in vitro for all of our inhibitor designs. However, we see a difference in the seeding inhibition of D1, D1b, and D1d when added to fibrils of Aβ or tau. Whereas all inhibitors affect tau monomer aggregation similarly, D1b is much more effective than the other inhibitors on fibril induced seeding. Additionally, inhibitors D1 and D1d mitigate Aβ seeding of tau aggregation at 20μM, whereas Tau induced seeding is not reduced at this concentration, suggesting that this is due to interaction with Aβ fibrils and not a direct inhibition of tau aggregation.

6) The paper concludes with the idea that "a direct interaction between the Aβ core and the amyloid-prone regions of tau facilitates cross-seeding". Can the authors be more specific about what this interaction is and how it facilitates cross-seeding, and about how their data support this idea? Do tau fibrils grow from the ends of Aβ fibrils? Or on the sides of Aβ fibrils? Or is a non-fibrillar state of Aβ important? Or perhaps a non-parallel structure that exists only at the ends of Aβ fibrils? Or perhaps tau oligomers form on the ends or sides of Aβ fibrils, and then tau fibrils nucleate in these tau oligomers?

We have not deduced the exact mechanism for cross-seeding, and cannot conclusively state the interaction that facilitates cross-seeding. Based on the results from our mutagenesis experiments, we have included a model as to how seeding could occur from sides of Aβ fibrils as Figure 6—figure supplement 3.

7) The work does not provide direct evidence that a common surface or motif exists in Aβ, as was found for tau. This could have been done simply by mutating Aβ, in the way that tau was mutated in this study and repeating the Ab seeding study of tau.

We thank the reviewers for this suggestion. We agree that we did not provide sufficient experimental evidence to provide a common surface in both Aβ and tau. We have included K18 biosensor seeding data and analysis for two Aβ mutants, Aβ L15R/F17R, and K16A/V18A/E22A in the text starting in subsection “Inhibitors reduce seeding of tau by aggregated Aβ_1-42_”, and in Figure 5—figure supplement 1. Residue F20 was not altered, as it has been shown both buried and surface accessible in full length structures. We found that fibrils of the L15R/F17R seed tau, while fibrils of K16A/V18A/E22A do not. Furthermore, seeding from fibrils of L15R/F17R is reduced upon addition of D1b. Together, we interpret this as indication that residues 16/18/22 are important for seeding and these residues are blocked by D1b,

We have also included the Ab_16-26_D23N segment in our in vitro seeding experiment (Figure 5A) of tau K18+, and found that it can seed, but less efficiently than K18 or Aβ42 fibrils.

8) The conclusions of this study are based on the development of peptide inhibitors that the authors demonstrate affect the formation of Aβ fibrils, with concomitant effects on toxicity and seeding. However, there is no assessment of the concentration of monomer or oligomers that remain in solution: these would be expected to increase as a consequence of impaired fibrillation. Such data would help to understand the mode of action of the presented inhibitors.

We agree that an assessment of species could help understand the mode of action of the presented inhibitors. When monomer is treated with inhibitor we see a reduction of A11 detected species and toxicity from the sample, implying oligomer formation is reduced. We have also added a line in the text (subsection “Reduction of toxicity by designed inhibitors is explained by a reduction of _Aβ1-42_ aggregation”) stating that we have not determined what oligomeric assemblies are being reduced by addition of our inhibitors.

9) Why is the crystalline arrangement of Aβ16-26 peptide described in this work considered to be a fibril? It is no different from previous steric zippers that are understood to be in a crystalline arrangement, such as those already described by the same group as steric zippers Class 7.

We have amended the text to call it a crystalline, fibril-like arrangement. Although crystalline, these structures are in fact fibrils by all criteria, except that they have no twist. We and others term these structures amyloid-like fibrils. They form elongated structures that can grow to contain infinite copies of the monomer.

10) Subsection “Efficacy of inhibitors of Aβ aggregation designed against Aβ 16-26 D23N”: please specify how IC50 was estimated.

We have amended the figure legend and Materials and methods sections to state how the IC50 estimate was calculated.

11) Subsection “Reduction of toxicity by designed inhibitors is explained by a reduction of Aβ_1-42_ aggregation”: The use of negative stain TEM to demonstrate the absence of fibrils is prone to over-interpretation. A more quantitative method, such as fibril yield (or soluble species yield) by centrifugation would strengthen the authors' argument.

We thank the reviewers for this point and agree that negative stain TEM can be over-interpreted in determining a presence or absence of fibrils. However, we are using this technique as a second method to validate our ThT results showing presence or absence of fibril formation. As this test is already a second verification, we do not feel that it is necessary to further validate our result.

12) A related issue is that there is a general lack of characterization of which particles are induced by interaction with inhibitor. DLS data would help to make the point regarding which species (monomers, oligomers, small fibrils) the inhibitor is binding to.

From our cellular and biochemical assays, the inhibitors appear promiscuous and could be binding to many forms of Aβ. The limited solubility and self-aggregation of the inhibitors has made the characterization of what distinct Aβ particles it binds to challenging. We have done our best to characterize the inhibitors’ effects on the various species (monomer, oligomer, fibril) despite not deducing the specific assemblies to which they bind.

13) A better use of Kd value for D1d inhibitor binding to Aβ fibrils would be to compare it with the Kd of D1b peptide which is less active than D1d. Also, there are other experimental methodologies such as microscale thermophoresis that have been reported to cope better with self-aggregation of the analytes that the authors claim to disturb their SPR based binding experiments.

We thank the authors for this comment. We agree that a comparison with inhibitor D1b would be informative, D1b was not soluble enough for this experiment. We attempted to measure binding using ITC, but solubility and self-aggregation of the inhibitors continues to disrupt our experiments. We are working to add solubility tags to our inhibitors, which could help with some of the difficulties in measurements.

14) Subsection “Inhibitors reduce seeding of tau by aggregated Aβ_1-42_”: the authors claim that in their experiments, fibrils of Aβ_1-42_ seed tau aggregation as efficiently as fibrils from tau K18. Yet the results don't show a clear reduction on Tlag (Figure 5A) that is expected and clearly observed in the Aβ_1-42_ self-seeding (Figure 5—figure supplement 1, panel B)

We thank the reviewers for this observation, and agree that the results of seeding of full length tau40 were ambiguous. We have replaced the current experiment in Figure 5A with seeding by fibrils of K18+. This experiment in the revised manuscript does not require the addition of heparin, which could influence the seeding from added fibrils of the tau monomer. We see a clear reduction of lag times seeding with K18, Aβ42 and Aβ16-26D23N. Our original Figure 5A is now included as Figure 5—figure supplement 1B, and we have included t_1/2_ calculations in the figure legend, showing a small seeding effect on the lag time of tau40 aggregation.

15) Subsection “Inhibitors reduce seeding of tau by aggregated Aβ_1-42_”: the authors imply that fibrils from hIAPP, TDP43, α-synuclein, and TTR are morphologically similar. This is incorrect.

Thank you for this point, we have amended the text.

16) Subsection “Inhibitors reduce seeding of tau by aggregated Aβ_1-42_”: the observed inhibition of Aβ_1-42_ fibril seeding of tau aggregation seems not to be dose dependent, except for inhibitor D1b. More experimental points in the dose response to inhibitors should be provided (Figure 5D).

We agree that more data points could be used to determine dose dependency and IC50s of the less effective inhibitors. However, we do not feel that inclusion of these additional data points would change our conclusion that D1b is more effective that the other inhibitors.

*17) Subsection “Inhibitors reduce tau aggregation and seeding”: control LC peptide (L-amino acid version of D1 peptide) does increase aggregation* in vitro *(reduced Tlag in ThT) and in cellular assays (increased number of seeded aggregates in tau-K18 biosensor cells). Can the authors comment how the stereochemistry of the peptides influences aggregation induction/inhibition?*

It could be that the capping interaction of the D-peptide is out of register, translated by ½ a bond distance perpendicular to the fibril axis, in order to maximize H-bonding with the amide backbone. This translation causes steric overlap between the side chains and incoming β sheets, which is the proposed mechanism of inhibition that is exploited by the D-peptide. For the L peptide, since H-bonding is satisfied by in register stacking, no steric overlap would be expected between the R groups of the L peptide, and incoming β sheets. Thus, it should not inhibit. In fact, we see some level of increased aggregation by LC, as the reviewer points out. It could be that the manufactured interaction of R groups in LC with the native Aβ sequence happen to be more hospitable to the native Aβ chain, than the homomeric stacking interactions that are native to the uninhibited fibril.

18) Discussion paragraph four: in this work there is no characterization of the aggregation pathways of Aβ and tau. It thus is inappropriate to suggest that both molecules share a common aggregation pathway based on responsiveness to a peptide inhibitor.

We agree with the reviewers that we showed no characterization of the aggregation pathways of Aβ and tau, and have removed this from the text.